# Quantifying and categorising the animal welfare impacts caused by biological invasions

Thomas Evans [1,2,3] & Michael Mendl [2]

Biological invasions cause animal suffering, but few studies assess these welfare impacts, and hence understanding of them is limited. We present a framework which can be used to identify relative changes to the welfare of an individual animal caused by biological invasions. We use it to assess the welfare impacts of bird and ant invasions. These impacts are a global phenomenon affecting native and introduced animals. Nevertheless, some introduced ant species cause severe impacts wherever they occur, whereas introduced birds do not. Impacts are likely to be underreported, particularly those affecting introduced animals. Physical and behavioural evidence (e.g., injuries and repetitive preening by birds) is sufficient to identify many welfare impacts. Physiological evidence (e.g., changes in 'stress' hormones) is more scarce, and could provide useful additional information to help quantify impact severity. Published biodiversity impacts of biological invasions are an unmined resource that may be used to assess impacts.

Biological invasions can be severely harmful to native biodiversity[1]. Preventing the damage they cause is a global conservation priority; one of the 23 global targets for 2030 laid out in the Kunming-Montreal Global Biodiversity Framework (GBF) requires coordinated, international efforts to reduce their impacts[2]. However, another type of impact caused by biological invasions that has not been widely considered, and by no means managed, is their adverse effects on the welfare (physical and mental state) of native and introduced animals[3,4].

The scientific study of animal welfare has been developing since the 1960s (e.g., Dawkins, 1980[5]; Fraser and Broom, 1996[6]; Mendl et al. 2017[3]; Appleby et al. 2018[7]; Mason et al. in press[8]), and there is growing consensus that the ability to consciously experience emotions and sensations (sentience) is a key determinant of animal welfare[9]. Whilst we cannot be certain whether and which non-human species are sentient, it is increasingly accepted, based on behavioural, cognitive and neuroscientific evidence, that mammals and birds, and likely other vertebrates and some invertebrates, have this capacity[10]—they are sentient beings[11] and treated as such in legislation[12]. Hence, in threatening or harmful situations, animals are likely to experience suffering through negative affective (emotional) states[13], including discrete

states such as anxiety, pain and fear[14], as well as potentially detrimental changes to their physical state.

Biological invasions can cause animal welfare impacts in many ways. For example, native birds on the Galápagos Islands are parasitised by the avian vampire fly (*Philornis downsi*) which was accidentally introduced to the archipelago in the mid-1900s. After hatching, fly larvae move to the ear canals of nestling birds to feed on their blood and keratin, causing swelling of the nostril area. Later stage larvae feed on the nestlings' bodies, creating open wounds under their wings and on their abdomens, legs and backs. These injuries often result in the death of the nestlings; those that survive are affected by permanent deformities, for example, adult birds are sometimes unable to vocalise effectively because they have unusually shaped nares (nostrils)[15,16].

Whilst much research is dedicated to the biodiversity impacts of biological invasions[17,18], less is dedicated to their animal welfare impacts. In illustration, there are many published reviews of the biodiversity impacts caused by different taxonomic groups of introduced species (e.g., cats[19], reptiles and amphibians[20]), but no reviews that explicitly assess their animal welfare impacts. As the biodiversity impacts of biological invasions affect the fitness of individual native

[1]Institute of Biology, Freie Universität Berlin, Berlin, Germany. [2]Bristol Veterinary School, University of Bristol, Langford, United Kingdom. [3]Leibniz Institute of Freshwater Ecology and Inland Fisheries (IGB), Berlin, Germany. ✉ e-mail: thomas.evans@fu-berlin.de

animals, sometimes causing injury, illness, physical deformities and death[15], they often have welfare impacts. Yet these are rarely described, the focus instead being on how severely biological invasions threaten native species with extinction. Nevertheless, there is growing interest in wild animal welfare[21], and while research on this topic tends to focus on the direct impacts of human activities on wildlife, it is timely to broaden the scope to include welfare impacts resulting from interactions between non-human organisms that are caused by biological invasions.

Frameworks are useful for structuring large-scale data, facilitating analysis and the development of hypotheses and theory[22]. The Environmental Impact Classification for Alien Taxa (EICAT) framework[23] can be used to quantify by severity and categorise by type the negative impacts that introduced species have on native species (i.e., impacts that affect the survival of native species—their ability to reproduce and sustain viable populations). For example, EICAT has been used to quantify and categorise the negative impacts of introduced birds[24] and ants[25] on native species. The dataset created by the introduced bird EICAT assessment was subsequently used for studies that improved understanding of their impacts, including the identification of factors that influence their severity[26] and the factors that cause native birds to be vulnerable to them[27]. These studies may inform biosecurity measures to prevent future invasions by the most harmful introduced bird species and to protect the native species most vulnerable to their impacts.

Here, we describe and test a framework which can be used to assess the animal welfare impacts caused by biological invasions—the Animal Welfare Impact Classification for Invasion Science (AWICIS). This framework is needed because the animal welfare impacts of biological invasions are a different type of impact to the biodiversity impacts of biological invasions (as assessed using the EICAT framework)[23,28]—animal welfare impacts affect the mental and physical state of individual animals[4]; biodiversity impacts affect the survival of entire species (their ability to reproduce and maintain viable populations). Furthermore, although the mechanisms through which biological invasions cause animal welfare and biodiversity impacts may often be the same (e.g., predation of a native species by an introduced species, which can cause declines in the population of the native species or even its extinction[29], and also affect the welfare of an individual of that species, which may suffer physically and emotionally whilst being preyed on[30]), the severity of the animal welfare and biodiversity impacts caused through the same mechanism may not be congruent. For example, introduced parasites may not cause species extinctions (a severe biodiversity impact), because as a species' population declines as a result of parasitism, transmission of the parasite is reduced to low densities[31]—however, introduced parasites can cause severe suffering of individual host animals[15]. Therefore, AWICIS may provide information on the severity of animal welfare impacts caused by biological invasions that cannot be obtained using existing frameworks such as EICAT. Finally, a dedicated animal welfare framework is required because biological invasions can affect the welfare of both native and introduced species, but existing frameworks that assess the impacts of biological invasions on non-human organisms (including EICAT[23,28]) have been designed solely to assess impacts on native species.

## Results
### AWICIS summary
A guidance document that justifies the framework's scope, describes its components and explains how it should be used, is provided in Supplementary Note 1. This document should be read prior to use of the framework—a summary of the main aspects of AWICIS is provided here.

AWICIS should only be used to assess the animal welfare impacts of biological invasions. Animal welfare is defined as being 'the physical and mental state of an animal in relation to the conditions in which it lives and dies'[4]. A separate (but related) framework, EICAT[23,28], assesses the biodiversity impacts of biological invasions (impacts affecting the survival of native species—their ability to reproduce and maintain viable populations). AWICIS does not assess impacts affecting the survival of species. Indeed, AWICIS assesses welfare impacts on individual animals (not species). These individuals may be native or introduced, wild or domesticated.

Animal sentience is 'the ability to have physical and emotional experiences, which matter to the animal, and which can be positive and negative'[32]. AWICIS can be used to assess the impacts of biological invasions on the welfare (physical and mental state) of both vertebrate and invertebrate species. However, when considering mental states to be the key determinant of welfare, inferences about such states can be made most strongly for animals that are protected under the Animal Welfare (Sentience) Act 2022[12] and assumed, at least in UK legislation, to be sentient. Currently, these are non-human vertebrates, cephalopod molluscs and decapod crustaceans.

Under AWICIS, the animal welfare impacts caused by a biological invasion are categorised by their mechanism (the way they cause suffering) (Table 1). AWICIS adopts 11 of the mechanisms identified for the EICAT framework, which assesses the impacts of introduced species on biodiversity[23,28]. AWICIS adopts these impact mechanisms because it is appropriate to do so—mechanisms that adversely affect the survival of native species (biodiversity impacts) can also adversely affect the physical and mental state of individual animals (animal welfare impacts). For example, transmission of an introduced disease may cause declining populations of a native species, but it can also cause physical injury and increase stress and anxiety in individual animals. The descriptions of these EICAT mechanisms have been amended slightly for AWICIS to ensure they are appropriate for the assessment of animal welfare impacts (see Supplementary Note 1, Section 2 for details). The AWICIS impact mechanisms are associated with domains of animal welfare as described in the published 2020 Five Domains Model[33], enabling links between mechanisms and domains to be identified (see Supplementary Note 1, Section 2). These domains are: (1) nutrition, (2) physical environment, (3) health, (4) behavioural interactions, and (5) mental state.

Animal welfare impacts are also quantified by their severity (the level of suffering they cause animals) (Supplementary Note 1, Section 3). To be of practical use, frameworks should provide a simple method to efficiently assess the impacts of biological invasions[22]. Hence AWICIS does not attempt to compare the severity of different types of welfare impacts caused by biological invasions (e.g., the effects of predation compared to the effects of a disease), nor does it assess severity across different types of animals (e.g., welfare impacts affecting a bird compared to those affecting a mammal)—this would be challenging, as different animals experience effects in different ways[34]. Rather, AWICIS identifies relative changes to the welfare of an individual animal that are associated with a specific type of impact caused by a biological invasion (e.g., predation). A lack of available data on impacts can be an impediment to the use of frameworks for invasion science. For example, a framework to assess the socio-economic impacts of introduced species on human well-being requires data on how these introduced species affect the ability of people to carry out their daily activities[35]. However, these data are sometimes unavailable for introduced species (as shown for introduced birds[36]). By focusing on relative changes to the welfare of individuals resulting from biological invasions, AWICIS provides a straightforward method that makes use of the many published studies on the biodiversity impacts of biological invasions, which include information that may be used to assess animal welfare impacts. As such, AWICIS adopts a pragmatic approach to impact assessment that helps to overcome potential issues associated with a lack of data on animal welfare impacts.

**Table 1 | AWICIS impact mechanisms**

| Impact mechanism | Description of welfare impacts on an individual animal caused by interactions with other species (plants, fungi, animals etc.) as a result of biological invasions |
| --- | --- |
| (1) Competition | Competition for resources (e.g., food, refuge) between a species and an animal |
| (2) Predation | A species attacks / kills / eats an animal |
| (3) Hybridisation | Hybridisation between two animals |
| (4) Disease transmission | A species transmits a disease to an animal |
| (5) Parasitism | A species parasitises an animal |
| (6) Poisoning / Toxicity | A species poisons an animal |
| (7) Disturbance | A species disturbs an animal (e.g., through loud vocalisations, but not through other mechanisms identified in this table, such as competition or predation) |
| (8) Chemical changes to ecosystem | A species changes chemical characteristics of the environment (e.g., pH, nutrient levels) which results in welfare impacts on an animal (e.g., biological invasions can create anoxic conditions in river systems, leading to the suffocation of fish[69]) |
| (9) Structural changes to ecosystem | A species changes the structure of the environment (e.g., species presence, abundance, diversity, complexity, spatial distribution, number of trophic levels) which results in welfare impacts on an animal (e.g., flammable and invasive introduced plants can change the structure of vegetation communities at the landscape scale, which are dominated by flammable plants, increasing the size of wildfires and the number of individual animals that they kill[70]) |
| (10) Physical changes to ecosystem | A species changes physical characteristics of the environment (e.g., thermal regimes, light regimes, water availability) which results in welfare impacts on an animal (e.g., introduced plants can alter thermal regimes of habitats, with lower and less variable temperatures reducing habitat suitability for native ectotherms[71]) |
| (11) Impacts through interactions with other species | A species interacts with another species resulting in welfare impacts on another animal e.g., introduced feral pigs (*Sus scrofa*) on Santa Cruz Island provided a food source for golden eagles (*Aquila chrysaetos*) which increased the eagle population size, and in turn the frequency of eagle predation on island foxes (*Urocyon littoralis*)[72] |

The impact mechanisms described in this table are adopted (and amended) from a published framework developed to assess the impacts of introduced species on native biodiversity—the Environmental Impact Classification for Alien Taxa (EICAT)[23,28].

**Table 2 | AWICIS impact severity categories**

| Impact severity categories | Description—the biological invasion causes: |
| --- | --- |
| (i) | negligible welfare impacts on an individual animal |
| (ii) | welfare impacts that result in short-term suffering similar to that affecting the individual animal in the absence of biological invasions |
| (iii) | welfare impacts that result in prolonged suffering similar to that affecting the individual animal in the absence of biological invasions |
| (iv) | welfare impacts that result in short-term suffering that the individual animal is not affected by in the absence of biological invasions |
| (v) | welfare impacts that result in prolonged suffering that the individual animal is not affected by in the absence of biological invasions |

AWICIS adopts the same structure (five impact severity categories) used for the EICAT framework[23,28] (Table 2). When assessing severity, AWICIS provides guidance to determine whether impacts are short-term or prolonged (Supplementary Note 1, Section 3). This enables distinctions to be made between level (ii) (short-term) and level (iii) (prolonged) impacts, and between level (iv) (short-term) and level (v) (prolonged) impacts (Supplementary Note 1, Table S2). AWICIS stipulates an approximate duration of < 1 hour for short-term impacts, with any impacts lasting longer than 1 hour being categorised as prolonged. Intermittent impacts that occur over long periods (> 1 hour) but not constantly (e.g., competition impacts) should be categorised as prolonged.

To establish the severity of an impact, comparisons must be made of the impacts caused by biological invasions with those that occur in their absence. AWICIS includes protocols to ensure these comparisons are appropriate by restricting them to impacts caused through the same mechanism and by similar types of taxa. It also provides protocols to assess welfare impacts on native and introduced species, as there are differences in the ways that individuals from these two groups of animals may be affected (Supplementary Note 1, Section 3, Fig. S3). When establishing impact severity, AWICIS also requires consideration of changes to the prevalence of exiting welfare impacts (their spatial or taxonomic extent, or their frequency).

AWICIS requires evidence of welfare impacts in the form of physical, behavioural or physiological indicators (e.g., physical injuries,

lethargy resulting from the effects of a disease, and elevated stress hormones as a marker of the threat of predation, respectively) (see Supplementary Note 1, Section 4). A confidence category of 'high', 'medium' or 'low' is assigned to each AWICIS assessment by the assessor to indicate their level of confidence that the impact severity category they have assigned to an impact is correct (see Supplementary Note 1, Section 6).

**Testing the framework**

To assess the ability of research scientists to use the framework without prior knowledge of it, we carried out a test exercise in three stages, whereby three research scientists were asked to: (i) review the guidelines and assessment template and provide comments, (ii) complete five 'training' impact examples using the assessment template and provide comments on the process, and (iii) complete a further ten impact examples with no assistance from T.E. or M.M. Following the established process for EICAT assessments submitted to the IUCN EICAT Authority, these ten assessments were reviewed by T.E. and any issues / errors discussed with the assessors, who then revised the assessments. The test exercise is described in detail in Supplementary Note 2.

All three assessors completed the ten assessment examples. Expected impact severity scores were allocated to 26 of the 30 assessments, and expected impact mechanisms to 25 of the 30 assessments. Errors could be attributed to the fact that the assessors

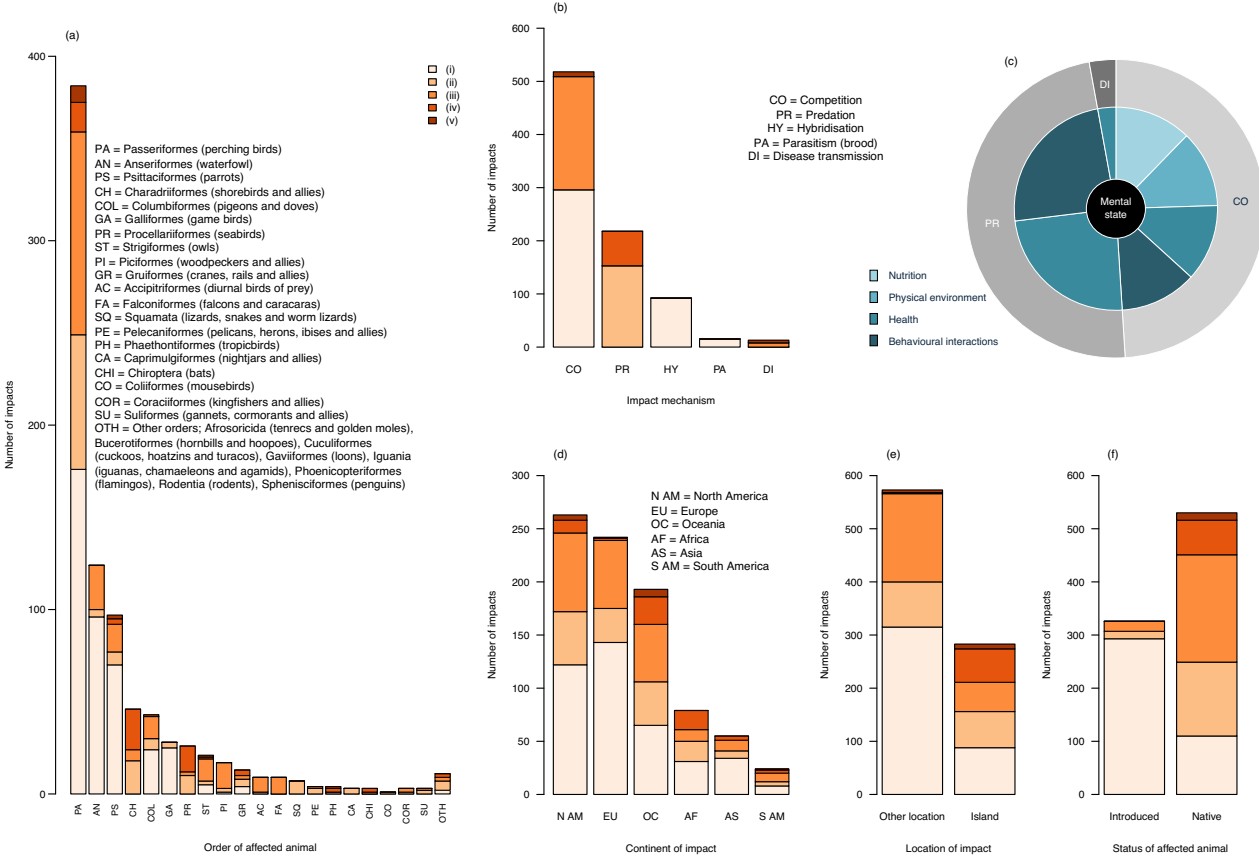

**Fig. 1 | The animal welfare impacts associated with bird invasions as assessed using AWICIS, including their number, severity, type, location, the animals they affect and their links with domains of animal welfare. a**, **b**—the number and severity of animal welfare impacts associated with bird invasions when distributed by order of affected animal and mechanism of impact, respectively; (**c**)—the proportion of animal welfare impacts when distributed by impact mechanism (outer circle) and by domain of animal welfare as defined in the published 2020 Five Domains Model[33] (inner circles) (hybridisation and parasitism had level (i) (negligible) impacts, so they do not affect domains of animal welfare and are not included in this figure); (**d**), (**e**), (**f**)—the number and severity of animal welfare impacts when distributed by continent of impact, location of impact (island or other location) and status of affected animal (introduced or native), respectively. Light orange to dark orange = increasing severity of animal welfare impacts associated with bird invasions. This colour scheme corresponds with the following five AWICIS impact severity categories: welfare impacts are (i) negligible, (ii) similar to those caused in the absence of biological invasions and short term, (iii) similar to those caused in the absence of biological invasions and prolonged, (iv) more severe than those caused in the absence of biological invasions and short-term, (v) more severe than those caused in the absence of biological invasions and prolonged.

were still learning how to use the framework (which they had no prior knowledge of), rather than because there were fundamental flaws with the framework. Indeed, the assessors all agreed with feedback provided by T.E.—there were no disagreements regarding assessment processes or outcomes. Confidence levels varied across assessors, but no assessments that were allocated a 'high' confidence by one assessor were allocated a 'low' confidence by another, or vice versa. Impact examples were either assessed as being of 'high' confidence by all assessors; 'high' confidence by some assessors and 'medium' confidence by others; or 'medium' confidence by some assessors and 'low' confidence by others.

### Applying the framework

To test the framework's utility in assessing different types of welfare impacts caused by widely separated taxa, we applied it to biological invasions associated with two groups of introduced species —birds and ants. In so doing, we classified the animal welfare impacts of biological invasions by their severity and type, and we used these data to investigate geographic and taxonomic patterns in their distribution.

### Impacts associated with bird invasions

Bird invasions affected the welfare of animals from three classes and 28 orders, including birds (23 orders), mammals (three orders) and reptiles (two orders) (Fig. 1a). Approximately 9% of all impacts were categorised as being 'more severe' ('more severe' impacts are level (iv) and (v) impacts grouped together to form one category—see Methods for details on grouping of impact severity categories for the analysis). Passeriformes (perching birds) sustained the most impacts (45%), followed by Anseriformes (waterfowl) (14%) and Psittaciformes (parrots) (11%). 'More severe' impacts were identified for individual animals from ten (37%) of these orders. Charadriiformes (shorebirds) and Procellariiformes (seabirds) were affected by a greater proportion of 'more severe' welfare impacts in comparison to other orders (Table 3, Test #1). Impacts affected the welfare of animals through five mechanisms—but mainly through two, being competition (61%) and predation (25%) (Fig. 1b). A greater proportion of predation impacts were 'more severe' when compared to impacts occurring through other mechanisms (Table 3, Test #2). Competition was linked with all four domains of animal welfare; predation was linked with Domain 3 (health) and Domain 4 (behavioural interactions) (Fig. 1c).

**Table 3 | A summary of the results of the six contingency table tests (unconditional exact tests) undertaken to identify broad patterns in the distribution of the animal welfare impacts caused by bird and ant invasions**

| | Contingency table test number and description | Result description | Statistic | Parameter | P value | Sample estimates | Table ref. |
|---|---|---|---|---|---|---|---|
| Bird invasions | #1. The distribution of welfare impact severity across order of affected animal. | Shorebirds and seabirds were affected by a greater proportion of 'more severe' welfare impacts when compared to other orders. | 57.8 | 4 | <0.001 | 0.17 | Table S6 |
| | #2. The distribution of welfare impact severity across impact mechanism. | A greater proportion of predation impacts were 'more severe' when compared to those caused by other mechanisms. Fewer competition impacts were 'more severe' when compared to those caused by other mechanisms. | 148.8 | 2 | <0.001 | 0.34 | Table S7 |
| | #3. The distribution of welfare impact severity across location of impact (island or other location). | A greater proportion of impacts on islands, and a smaller proportion of impacts at 'other locations' were 'more severe'. | 117.4 | 1 | <0.001 | 0.37 | Table S8 |
| Ant invasions | #4. The distribution of welfare impact severity across class of affected animal. | Not significant. | 0.33 | 1 | 0.57 | 0.05 | Table S9 |
| | #5. The distribution of welfare impact severity across impact mechanism. | A greater proportion of predation impacts were 'more severe' when compared to impacts caused by 'other mechanisms' (competition, disturbance and poisoning / toxicity impacts grouped together). | 15.34 | 1 | <0.001 | 0.37 | Table S10 |
| | #6. The distribution of welfare impact severity across location of impact (island or other location). | Not significant. | 0.3 | 1 | 0.59 | 0.05 | Table S11 |

All tests were two-sided and carried out in R using the FunChisq package[65]. For the complete contingency tables, see Supplementary Note 3 (table references are provided in the final column of this table). 'Less severe' impacts = level (i), (ii) and (iii) impacts grouped together to form one category; 'more severe' impacts = level (iv) and (v) impacts grouped together to form one category.

Indicators of animal welfare impacts were either behavioural (63%) or physical (37%). Behavioural indicators included: (a) observations of aggressive interactions such as predation of one bird by another, or aggressive competition for food between waterfowl species; (b) abandonment of nest cavities by Eurasian nuthatches (*Sitta europaea*) due to aggressive interactions with rose-ringed parakeets (*Alexandrinus krameri*), and (c) listless American goldfinches (*Spinus tristis*), and lethargic Hawaii amakihi (*Chlorodrepanis virens*) and Galapagos doves (*Zenaida galapagoensis*) with a loss of balance and breathing difficulty—all three bird species with loss of appetite and weight loss due to the effects of various diseases. Physical indicators included: (a) dead animals, such as the remains of introduced rose-ringed parakeets in the pellets of native long-eared owls (*Asio otus*), and the carcasses of native seabirds killed by introduced barn owls (*Tyto alba*); (b) bird nests and eggs destroyed by common mynas (*Acridotheres tristis*) including nests of grey warblers (*Gerygone igata*); (c) injured animals such as anaemic Hawaii amakihi, blind American goldfinches with swollen, red, crusty eyes sockets, glassy eyes and respiratory-tract infections, and Galapagos doves with mouth lesions, inflamed and ulcerated mucosal surfaces lining the respiratory-tract, necrotic masses, abscessation of the oropharynx and diarrhoea.

Impacts occurred across many regions of the world, although they were often reported in high-income regions, particularly North America, Europe and Oceania (Fig. 1d; Fig. 2); many impacts were reported on islands (33% of all impacts) (Fig. 1e; Fig. 2). A greater proportion of impacts on islands, and a smaller proportion at other locations, were 'more 'severe' (Table 3, Test #3). Most welfare impacts were sustained by native animals (62%); no introduced animals were reported to be affected by 'more severe' impacts (Fig. 1f).

Half of all impacts were assessed as being of 'high' confidence—these impacts tended to be observed, predation impacts causing physical injuries; 48% of all impacts were of 'medium' confidence—often due to uncertainty over the duration of a predation impact (short-term or prolonged) and hence the level of suffering involved; and 2% were of 'low' confidence— all being brood parasitism impacts categorised as negligible (level (i)), but with uncertainty regarding possible fitness and energetic impacts on parenting birds that would require additional physiological research to identify.

### Impacts associated with ant invasions

Ant invasions affected the welfare of animals from six classes and 27 orders, including birds (14 orders), mammals (seven orders), reptiles (three orders), amphibians, crustaceans and ray-finned fishes (one order each) (Fig. 3a). Almost all impacts (92%) were categorised as being 'more severe'. Passeriformes (perching birds) sustained the most impacts (16%), followed by Squamata (lizards, snakes and worm lizards) (14%) and Testudines (turtles) (12%). 'More severe' impacts were identified for individual animals from all but three of the 27 affected orders (Chiroptera (bats), Columbiformes (pigeons and doves) and Strigiformes (owls)). Impact severity was randomly distributed across class of affected animal (Table 3, Test #4). Impacts affected the welfare of animals through four mechanisms—but mainly through predation (85%) (Fig. 3b). A greater proportion of predation impacts were 'more severe' when compared to impacts occurring through 'other mechanisms' (competition, disturbance and poisoning / toxicity impacts grouped together) (Table 3, Test #5). Four mechanisms were linked with domains of animal welfare, with the most prevalent mechanism, predation, being linked with two domains (Domain 3, health and Domain 4, behavioural interactions) (Fig. 3c). The earliest report of a welfare impact was from 1948 (introduced ants causing holes in the webbed feet of wedge-tailed shearwaters (*Ardenna pacifica*) in Hawaii).

Indicators were mainly physical (86%) though also behavioural (12%)—few were physiological (2%). Physical indicators included: (a) dead animals such as fledgling snapping turtles (*Chelydra serpentina*)

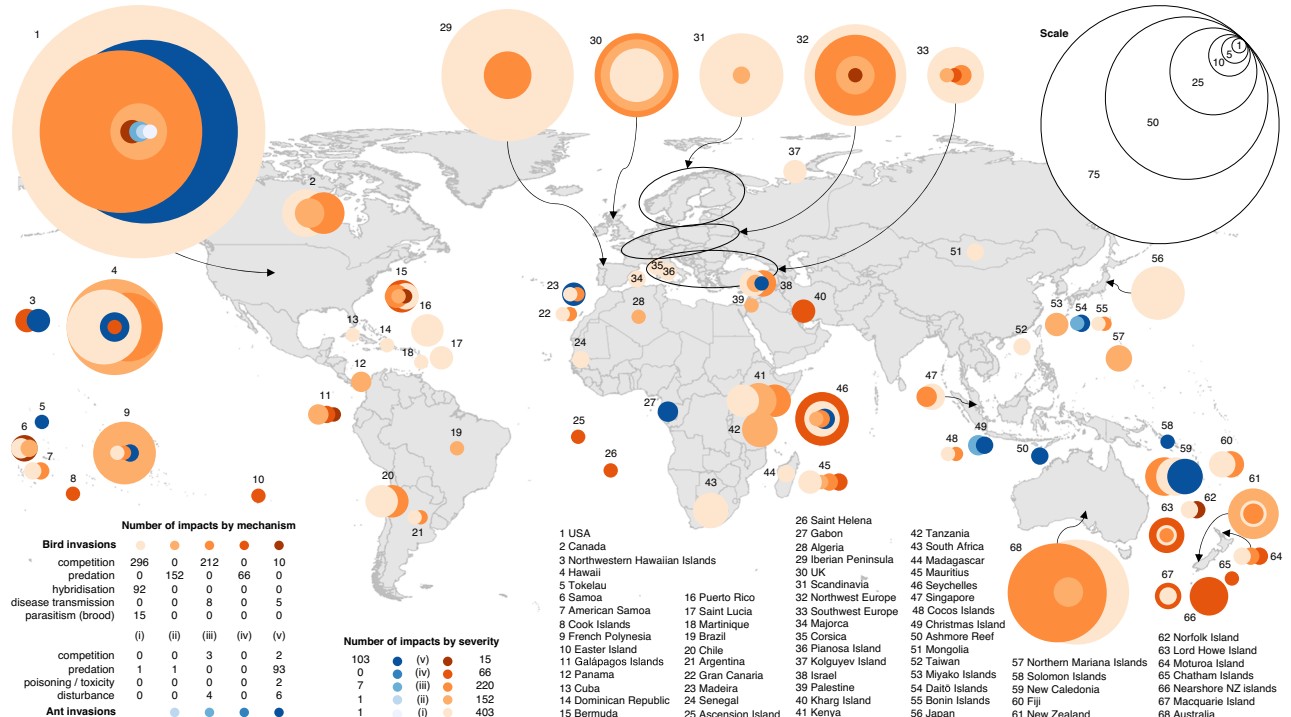

**Number of impacts by mechanism**

**Bird invasions**

|  | (i) | (ii) | (iii) | (iv) | (v) |
|---|---|---|---|---|---|
| competition | 296 | 0 | 212 | 0 | 10 |
| predation | 0 | 152 | 0 | 66 | 0 |
| hybridisation | 92 | 0 | 0 | 0 | 0 |
| disease transmission | 0 | 0 | 8 | 0 | 5 |
| parasitism (brood) | 15 | 0 | 0 | 0 | 0 |

|  | (i) | (ii) | (iii) | (iv) | (v) |
|---|---|---|---|---|---|
| competition | 0 | 0 | 3 | 0 | 2 |
| predation | 1 | 1 | 0 | 0 | 93 |
| poisoning / toxicity | 0 | 0 | 0 | 0 | 2 |
| disturbance | 0 | 0 | 4 | 0 | 6 |

**Ant invasions**

**Number of impacts by severity**

| 103 | (v) | 15 |
|---|---|---|
| 0 | (iv) | 66 |
| 7 | (iii) | 220 |
| 1 | (ii) | 152 |
| 1 | (i) | 403 |

1 USA
2 Canada
3 Northwestern Hawaiian Islands
4 Hawaii
5 Tokelau
6 Samoa
7 American Samoa
8 Cook Islands
9 French Polynesia
10 Easter Island
11 Galápagos Islands
12 Panama
13 Cuba
14 Dominican Republic
15 Bermuda
16 Puerto Rico
17 Saint Lucia
18 Martinique
19 Brazil
20 Chile
21 Argentina
22 Gran Canaria
23 Madeira
24 Senegal
25 Ascension Island
26 Saint Helena
27 Gabon
28 Algeria
29 Iberian Peninsula
30 UK
31 Scandinavia
32 Northwest Europe
33 Southwest Europe
34 Majorca
35 Corsica
36 Pianosa Island
37 Kolguyev Island
38 Israel
39 Palestine
40 Kharg Island
41 Kenya
42 Tanzania
43 South Africa
44 Madagascar
45 Mauritius
46 Seychelles
47 Singapore
48 Cocos Islands
49 Christmas Island
50 Ashmore Reef
51 Mongolia
52 Taiwan
53 Miyako Islands
54 Daitō Islands
55 Bonin Islands
56 Japan
57 Northern Mariana Islands
58 Solomon Islands
59 New Caledonia
60 Fiji
61 New Zealand
62 Norfolk Island
63 Lord Howe Island
64 Moturoa Island
65 Chatham Islands
66 Nearshore NZ islands
67 Macquarie Island
68 Australia

**Fig. 2 | The global distribution of animal welfare impacts associated with bird and ant invasions as assessed using AWICIS, including their number, severity and type.** 1–68 = broad location of impact. Light orange to dark orange = increasing severity of animal welfare impacts associated with bird invasions; light blue to dark blue = increasing severity of animal welfare impacts associated with ant invasions. These colour schemes correspond with the following five AWICIS impact severity categories: welfare impacts are (i) negligible, (ii) similar to those caused in the absence of biological invasions and short term, (iii) similar to those caused in the absence of biological invasions and prolonged, (iv) more severe than those caused in the absence of biological invasions and short-term, (v) more severe than those caused in the absence of biological invasions and prolonged. The map in this figure was produced in R using Natural Earth (www.naturalearthdata.com).

and small mammals caught in traps (for capture-release surveys) that were unable to avoid predation; (b) injured animals such as hatchling American alligators with swollen heads and bodies, Daito White-eyes (*Zosterops japonicus daitoensis*) with corneal inflammation and puss-filled nostrils, wedge-tailed shearwaters and red-tailed tropicbirds (*Phaethon rubricauda*) with bill and eye irritation, blind and maimed Christmas Island red crabs (*Gecarcoidea natalis*), and domestic cats (*Felis catus*), domestic dogs (*Canus familiaris*), leopards (*Panthera pardus*) and African savanna elephants (*Loxodonta Africana*) with Florida keratopathy (TK) (a health condition characterised by cloudy eyes); (c) animals with physical abnormalities such as wedge-tailed shearwaters with missing phalanges (toes), asymmetrical and missing nares (nostrils), discoloured and malformed bills, asymmetrical eyes and partial or complete eye occlusion due to overgrowth of skin or swelling; and (d) malnourished animals including fledgling eastern bluebirds (*Sialia sialis*) and green turtles (*Chelonia mydas*). Behavioural indicators included: (a) vigorous avoidance and defensive behaviour in fledgling black-capped vireos (*Vireo atricapilla*); (b) repetitive movements such as sooty terns (*Onychoprion fuscatus*) shaking their feet and wedge-tailed shearwaters shaking their heads and excessively preening; (c) diminished resting behaviour by Christmas Island flying-foxes (*Pteropus natalis*); (d) abandonment of parenting activities, including diminished nest caring behaviour by broad-snouted caimans (*Caiman latirostris*), nest abandonment by red-tailed tropicbirds and Texas river cooters (*Pseudemys texana*), abandonment of parenting behaviour by eastern bluebirds (*Sialia sialis*) and erratic incubation behaviour by European herring gulls (*Larus argentatus*); (e) sneezing and abnormal breathing by wedge-tailed shearwaters; and (f) lethargy in wedge-tailed shearwaters and red-tailed tropicbirds. One article used a physiological indicator (elevated glucocorticoid (CORT) levels) to indicate stress in native fence lizards (*Sceloporus undulatus*) occupying areas invaded by introduced ants.

Impacts were distributed across several regions of the world including North America (71% of all impacts) (Figs. 2, 3d) and islands (36%) (Figs. 2, 3e). Impact severity was randomly distributed across broad geographic location (islands when compared to other locations) (Table 3, Test #6). Almost all reported welfare impacts affected native species (95%). No impacts affected wild, introduced animals, but six impacts (5%) affected domestic, introduced animals—domestic cats and domestic dogs (Fig. 3f).

Most impacts were assessed as being of 'high' confidence (76%)—these tended to be observed, predation impacts causing physical injuries; 9% were of 'medium' confidence—often due to uncertainty over the duration of an impact (short-term or prolonged) or uncertainty regarding the severity of the impact in comparison to those caused by native species in the absence of the biological invasion; and 15% were of 'low' confidence—often due to uncertainty regarding the taxonomic identify of the ant species or because these were inferred predation impacts (for example based on the conspicuous absence of native animals in regions invaded by introduced ants in comparison to uninvaded regions, and the reported predation impacts of introduced ants on similar animals elsewhere).

## Discussion

Biological invasions affect the welfare of both introduced and native animals in many ways, and in some cases the welfare impacts they cause are more severe than those that affect these animals in their absence[37]. Despite this, little research has been undertaken to explicitly identify and assess the animal welfare impacts of biological invasions. Frameworks can be used to bring order to large and disparate datasets, and with regard to biological invasions, they can be used to apply a

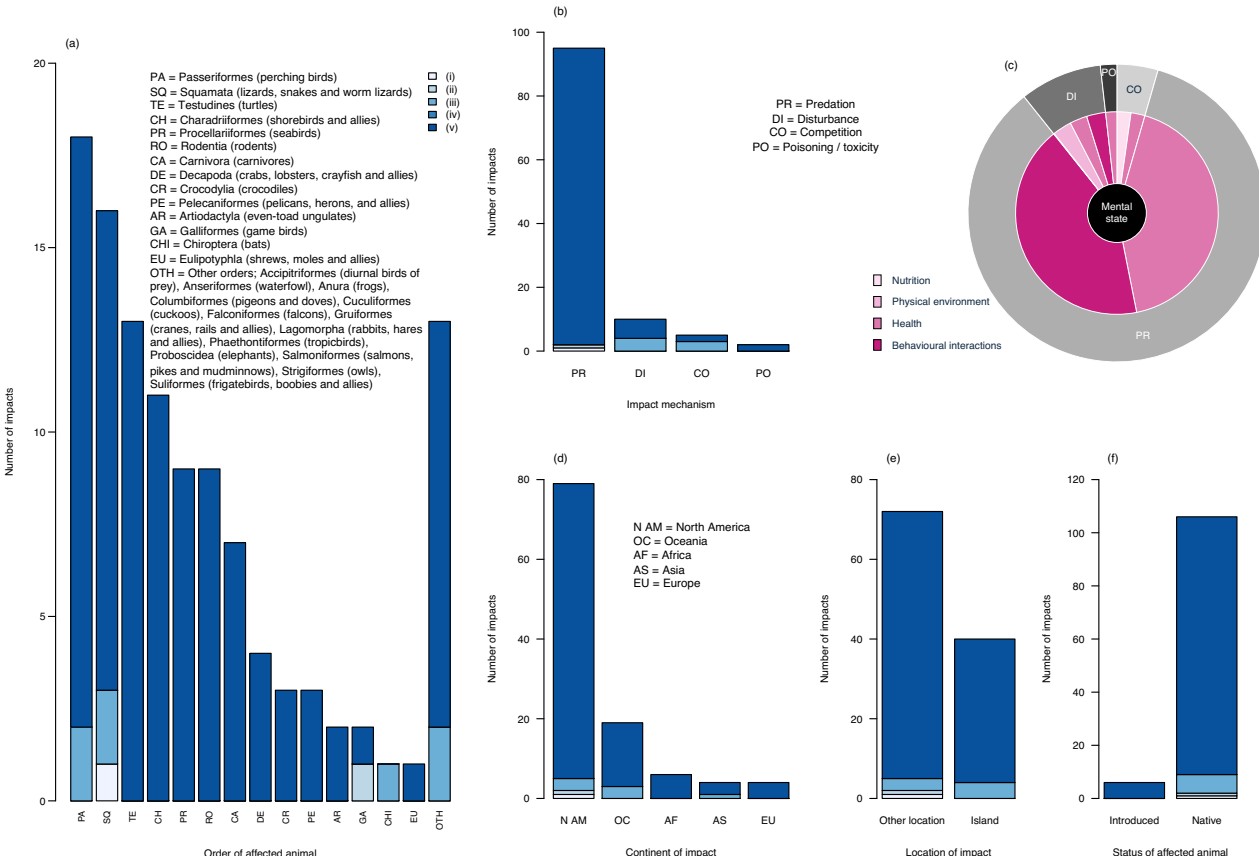

**Fig. 3 | The animal welfare impacts associated with ant invasions as assessed using AWICIS, including their number, severity, type, location, the animals they affect and their links with domains of animal welfare. a**, **b**—the number and severity of animal welfare impacts associated with ant invasions when distributed by order of affected animal and mechanism of impact, respectively; (**c**)—the proportion of animal welfare impacts when distributed by impact mechanism (outer circle) and by domain of animal welfare as defined in the published 2020 Five Domains Model[33] (inner circles); (**d**), (**e**), (**f**)—the number and severity of animal welfare impacts when distributed by continent of impact, location of impact (island or other location) and status of affected animal (introduced or native), respectively. All impacts on introduced animals affected domesticated species (cats and dogs). Light blue to dark blue = increasing severity of animal welfare impacts associated with ant invasions. This colour scheme corresponds with the following five AWICIS impact severity categories: welfare impacts are (i) negligible, (ii) similar to those caused in the absence of biological invasions and short term, (iii) similar to those caused in the absence of biological invasions and prolonged, (iv) more severe than those caused in the absence of biological invasions and short-term, (v) more severe than those caused in the absence of biological invasions and prolonged.

common structure from which to assess the severity and type of their impacts[22]. Here, we have described and tested a framework which can be used to quantify by severity and categorise by type the animal welfare impacts caused by biological invasions: the Animal Welfare Impact Classification for Invasion Science (AWICIS).

## Study limitations

By applying AWICIS to assess the animal welfare impacts associated with bird and ant invasions, we demonstrate that these welfare impacts are a global phenomenon, occurring across many regions of the world, including both high- and low-income regions, and many islands. Nevertheless, our study is unlikely to have identified the complete range of welfare impacts caused by bird and ant invasions. The biodiversity impacts of introduced species tend to be more frequently reported in high-income regions[38]—as our data on welfare impacts were taken from reports of biodiversity impacts, they are skewed towards these regions, particularly North America, Western Europe and Australasia. Furthermore, biodiversity research tends to focus on the more severe biodiversity impacts of introduced species[38], and therefore our dataset is also likely to be skewed toward the types of welfare impacts they are associated with. As such we are likely to lack data on welfare impacts in low-income regions, and also on the types

of welfare impacts associated with less severe biodiversity impacts. This issue of data availability is also apparent for published assessments of the biodiversity impacts of introduced species[1,39]. Hence, absence of evidence on impacts should not be considered as evidence of absence of impacts—just because we did not identify welfare impacts for certain taxa or regions does not mean they do not occur. The conclusions we draw in the following sections are based on the data that are available.

## Findings for bird invasions

Based on the data that are available, our results suggest that bird invasions rarely cause welfare impacts on animals that they do not already experience. This is because the reported welfare impacts of bird invasions tend to occur through two widespread mechanisms, competition and predation[24]—the processes associated with these mechanisms tend to be commonplace in ecological communities. Hence, the more severe animal welfare impacts associated with bird invasions tend to occur where there are few or no native bird species causing welfare impacts prior to the arrival of introduced birds. Such locations are often islands – for example, introduced raptors have been introduced to many islands with no resident raptor species in attempts to control introduced rats (*Rattus* spp.)[40]. On these islands,

introduced raptors prey on birds (for example, introduced Australian masked owls (*Tyto novaehollandiae*) prey on white turns (*Gygis alba*) on Lord Howe Island)[41]. This is why we found many of the more severe welfare impacts caused by bird invasions to affect shorebirds and seabirds. The biodiversity impacts of introduced birds tend to be more severe on islands[27], because they often affect endemic native species, and hence are more likely to cause species extinctions. Our results therefore suggest that there may be a link between the severity of the biodiversity impacts and animal welfare impacts of bird invasions. However, further refinement of AWICIS, followed by formal analyses using AWICIS and EICAT data would be required before any such link could be confirmed.

We identified two mechanisms associated with bird invasions that current data suggest have negligible impacts—hybridisation and brood parasitism. Nevertheless, with regard to hybridisation, it is plausible that some hybrid offspring may suffer from complications associated with developmental defects and morphological abnormalities, and are less fit and hence more vulnerable to impacts through competition, predation or disease transmission. For example, hybrids from crosses between two species of dwarf hamsters (*Phodopus campbelli* and *Phodopus sungorus*) (albeit both native) resulted in placental and embryonic overgrowth, severe developmental defects and maternal death[42]. With regard to brood parasitism, the negative biodiversity impacts of introduced brood parasites (e.g., shiny cowbirds (*Molothrus bonariensis*) affecting the nesting success of yellow-shouldered blackbirds (*Agelaius xanthomus*) in Puerto Rico)[43] may have welfare impacts on host animals, such as increased (and potentially stressful and fitness-threatening) energetic expenditure when rearing host chicks, which could be identified using physiological indicators.

### Findings for ant invasions

The data that are available suggest that a greater range of taxa are vulnerable to severe welfare impacts caused by ant invasions when compared to bird invasions. The ant invasions in our dataset also consistently cause severe welfare impacts wherever they occur, whereas the bird invasions rarely do. The severe welfare impacts of ant invasions identified in our study are mainly associated with one impact mechanism—predation. This common impact mechanism affects the welfare of many animals in the absence of biological invasions—hence in many cases predation by introduced species is unlikely to cause impacts on animals that they do not already experience (as we found for introduced birds). However, some introduced ant species use acid or venom to attack and subdue their prey (e.g., yellow crazy ants, *Anoplolepis gracilipes*), and because the quantity of acid or venom administered by an individual ant is small, it can take a long time for ants to kill an animal (indeed, the larger the animal the slower the death, and hence the greater the assumed suffering). An indicator of the suffering introduced ants cause other animals is the severe impacts they have on humans – the sting of the red imported fire ant (*Solenopsis invicta*) can kill an adult human and cause anaphylactic shock[44]. An example of an animal welfare impact caused by introduced ants includes a fledgling bull-headed shrike (*Lanius bucephalus*) attacked by yellow crazy ants on Minami-Daito Island, which suffered from corneal inflammation and died of its injuries after several days[45]. Predation by introduced ants has also been reported to cause stress and vigorous avoidance and defensive behaviour in animals, including fledgling black-capped vireos (*Vireo atricapilla*) attacked by red imported fire ants in their nests, which succumbed to their injuries over several hours[46]. Although this example is of impacts on nesting birds, impacts often occur at ground level, particularly to immobile, ground-nesting species—this is why we found many impacts affecting fledgling turtles, seabirds, shorebirds and crocodiles.

From the data that are available, these severe welfare impacts are not only restricted to islands where there are no native insect species that cause the same types of impacts. Indeed, many impacts caused by

the red imported fire ant were reported in mainland USA, where there are native fire ant species such as the tropical fire ant (*Solenopsis geminata*) that cause similar impacts. Nevertheless, the red imported fire ant is more aggressive than the tropical fire ant[47] and causes widespread welfare impacts on individual animals that would in all likelihood be unaffected in its absence[48,49]. Hence, our results do not reveal patterns in the severity of welfare impacts across different locations, nor across different introduced ant species or affected animals, that suggest tailored management actions may be appropriate. Indeed, the biodiversity impacts of introduced ants are often severe[50], their economic costs can be high[51] (and in comparison to birds[52]), they can kill people[44] and animals, and likely cause significant suffering in the process. Clearly, some introduced ant species are an all-round threat to 'One Health'—preventing their introduction is necessary to protect nature, the economy, human health and animal welfare (though not all—just 17 of > 500 ant species introduced to new environments are considered harmful to biodiversity[53]).

### Assessing welfare using indicators

Few animal welfare impacts were identified using physiological indicators, most likely because they are mainly used for studies that assess wild animal welfare[54], which are limited in number in comparison to studies on biodiversity impacts, which often include observations of an animal's physical condition and behaviour (the other types of indicators that can be used to assess welfare). Furthermore, physiological indicators require research techniques, such as the collection and analysis of samples, that are more complex and costly than the techniques required for physical and behavioural indicators (often just observations), which may also explain their absence in our dataset. For ant invasions, it may not be necessary to use physiological indicators, as their impacts tend to be evident in the physical condition and behaviour of affected animals. However, for more subtle impacts, particularly those associated with introduced birds such as competition, it may be useful to provide additional evidence regarding the intensity of likely welfare impacts by documenting physiological indicators (e.g., greater elevations in stress hormones in affected animals which may indicate greater challenges[55,56]). Indeed, we identified a severe animal welfare impact associated with changing behaviour of cavity-nesting, native Eurasian nuthatches when competing for nest cavities with cavity-nesting, introduced rose-ringed parakeets in Belgium. The native nuthatches abandoned their nest cavities—they do not do so when competing with other native bird species[57]. This suggests a level of stress and perceived threat that is not experienced in the absence of the introduced parakeets—physiological indicators could help to support this assumption[21]. Overall, many of the welfare impacts identified in this study (e.g. injury, disease, damaging behaviour, prolonged death) were relatively clear-cut in terms of their likely links to poor welfare. However, as discussed in the AWICIS guidelines (Supplementary Note 1), it will be important to ensure that conclusions from this framework are based on well-validated welfare indicators, and to acknowledge when there may be some uncertainty as to the actual welfare impact on the animal of interest (using the confidence categories of 'low', 'medium' and 'high').

### Future directions for research / policy implications

The examples of welfare impacts caused by biological invasions (other than birds and ants) provided in the AWICIS guidance documentation (Supplementary Note 1, Figs. S2 and S3) indicate that the animal welfare impacts of biological invasions are both caused by, and affect, a diverse range of taxa. Several impact examples are also provided in the AWICIS assessment template (Supplementary Note 1, Section 7). Taken together with the fact that our dataset is skewed toward high-income regions, it is likely that there are considerable gaps in our understanding of the taxonomic and spatial extent of the animal welfare impacts caused by biological invasions, respectively. AWICIS could be

used to improve this understanding—indeed, there are many published studies of the biodiversity impacts of biological invasions for a broad range of introduced taxa which may be used to identify and assess these impacts. However, our results indicate that whilst there may be plenty of evidence of impacts in the form of physical and behavioural indicators, there may be less evidence reported using physiological indicators, and hence some of the less apparent / visible welfare impacts caused by biological invasions may require additional empirical physiological research[21].

AWICIS is an indicator-based framework—future research that aims to assess the animal welfare impacts of biological invasions using the framework should collect data relevant to one or more of the three indicators described in the AWICIS guidelines (physical, behavioural and physiological indicators; Supplementary Note 1, Table S4). Future biodiversity research not concerned with welfare impacts but that collects data relevant to these indicators (e.g., observations of the physical appearance and behaviour of animals) may provide useful data used for AWICIS assessments—hence we encourage biodiversity researchers to take advantage of opportunities to gather data relevant to animal welfare when undertaking biodiversity field research. The data we obtained for this study mainly described impacts on native animals rather than introduced animals—indeed, impacts associated with ant invasions were only reported to affect native animals and a small number of domesticated animals, and yet introduced ants are likely to harm introduced animals in the same way that they do native animals. One of the reasons for this absence of data is that biodiversity research tends to focus on impacts affecting native species (not introduced species). Future studies reporting the biodiversity impacts of introduced ants (and biological invasions in general) could incorporate consideration of their animal welfare impacts on native and introduced animals, including assessments of their severity using AWICIS.

## Methods

The methods adopted for the test assessments undertaken by three research scientists are described in Supplementary Note 2.

We used AWICIS to assess the animal welfare impacts of bird and ant invasions. Birds have been introduced to novel environments for thousands of years[58]—introduced bird species are now established in most of the world's bioregions[59] including on many small islands[40]. The number of established introduced bird species continues to rise[1], and some are reported to have severe impacts on biodiversity[24]—for example, predation by introduced birds is likely to have contributed to the extinction of at least four native bird species on small islands[40]. Ants have also been introduced to many regions of the world—there are at least 520 introduced species that are established worldwide[53]. Three of these species are amongst the ten most widespread invasive insects[1]. Ants are often considered to be amongst the most damaging introduced species—17 ant species are known to have severe impacts on native biodiversity[53], including the yellow crazy ant which through predation has severely reduced the size of the red crab population on Christmas Island[60].

Because many different bird and ant species have been introduced globally[53,58], and their biodiversity impacts have been comprehensively studied[24,25,40,53], they provide a good opportunity to test AWICIS using global datasets of impacts caused by a diverse range of introduced species. We also chose ants and birds because they are taxonomically distinct, and we therefore expected them to have different types of welfare impacts. Hence, through this study we aim to demonstrate how AWICIS may be used to: (a) assess the welfare impacts caused by a complete taxonomic class of introduced species (birds), and (b) compare the animal welfare impacts associated with different taxonomic groups of introduced species (birds vs ants).

Introduced species have a global distribution[61], and research indicates that they interact with and affect native species in many ways—in some cases, their biodiversity impacts are severe, in other cases they are negligible[1]. As biodiversity impacts affect animal welfare, it is reasonable to assume that the severity and type of the welfare impacts associated with biological invasions will vary. We therefore adopt the following broad hypothesis: *The severity of the welfare impacts associated with biological invasions varies across (a) taxonomy of introduced species, (b) taxonomy of affected animals, (c) mechanism of welfare impact, and (d) location of welfare impact.* Similar, open-ended hypotheses have been used to identify broad patterns in the distribution of large ecological datasets[62], including those on the impacts of introduced species[24].

### Data

For bird invasions, we used two published global datasets on their biodiversity impacts[24,40]. The first dataset is a global assessment of the biodiversity impacts of introduced birds using EICAT[24] and the second is an assessment of their predation impacts on native birds on islands[40]. These datasets were gathered through online, desk-based literature reviews—together they form the most comprehensive dataset on the biodiversity impacts of introduced birds. For ant invasions, we undertook an online search to identify published articles describing their environmental impacts. Following Evans et al. (2016)[24] we used a series of search terms in a search string entered into Google Scholar: ("introduced species" OR "invasive species" OR "invasive alien species" OR "IAS" OR "alien" OR "non-native" OR "non-indigenous" OR "invasive" OR "pest" OR "feral" OR "exotic") AND ("ant" OR "formicidae"). We produced a long-list containing the first 200 results, reviewing the article abstracts to assess their relevance (i.e., studies describing biodiversity impacts associated with ant invasions). We then reviewed the reference lists of all the studies we selected, in order to identify any additional studies not identified through the online search, again selecting those describing the biodiversity impacts of ant invasions based on information provided in the abstracts. We repeated this process, reviewing the reference lists of all newly identified articles, to the point where no additional relevant articles were identified. Our search was informed by several 'review' articles, including global and regional reviews of the biodiversity impacts of introduced ants, and reviews of the impacts of specific introduced ant species. Together, these 'review' articles contained many references for published articles describing biodiversity impacts. The list of articles identified by the online search and the list of 'review' articles have been uploaded to Figshare[63]. We excluded information that was not peer-reviewed (e.g., many websites describing the impacts of introduced ants). We also excluded articles describing impacts caused by ant species in their native ranges, and articles that did not contain original research. We also excluded studies describing impacts that were identified using experiments carried out in artificial settings (laboratory or field experiments), although if a field experiment compared manipulated sites with unmanipulated sites where impacts occurred naturally, we included data on impacts occurring at unmanipulated sites. We also included studies that reported the impacts of fire ants on small mammals that had been trapped for mark and recapture schemes[64], as these impacts did not occur as a result of experiments. Examples of excluded articles have been uploaded to Figshare[63] (we have provided example references as an exhaustive list of excluded articles would be long).

### Analysis

For each published interaction identified in the literature, we used AWICIS to assess the welfare impacts on any affected animals (native and introduced, wild and domesticated), assigning to each an impact mechanism and an impact severity score of (i) – (v). As described in the AWICIS guidelines (Supplementary Note 1, Section 3.5), we compared the severity of the welfare impacts on an affected animal caused by a biological invasion with those caused by native species in the absence

of the biological invasion in the animal's native range (or in the country of impact for affected domesticated animals). To identify existing impacts caused by native species, we reviewed the native species assemblage, identifying impacts that would occur through the same mechanism as those caused by the biological invasion. As stipulated in the AWICIS guidelines, these impacts were inferred based on the native species present and their likely interactions. Also as stipulated in the AWICIS guidelines (Supplementary Note 1, Section 3.3), these impacts had to be caused by native species of the same class as those caused by the impacting species associated with the biological invasion (e.g., welfare impacts on an animal caused through predation by an introduced ant were compared to welfare impacts on the animal caused by predation by native insects (including native ants) in the affected animal's native range).

To identify variation in the severity of the animal welfare impacts in our dataset, we used contingency table tests (unconditional exact tests using the FunChisq package[65] in R[66]) to compare the actual and expected number of impacts that were 'less severe' and 'more severe' as distributed across different categories of interest (e.g., impact mechanisms). An impact was defined as being a description of a welfare impact on an individual animal occurring through a specific mechanism, as reported in a research article. Descriptions of the same types of impacts on many individuals of the same species were counted as a single impact. Where a research article described welfare impacts on individuals of different species, an impact was counted for each affected species. Impacts on individual animals of the same species reported in different studies were counted separately.

The categories of interest that we compared were: (a) order of affected animal, (b) mechanism of impact, and (c) location of impact (either an island ($< 20,000 \text{ km}^2$) or other location). We included islands as a category because the biodiversity impacts of introduced species, including birds, tend to be more severe on islands[27], and we aimed to establish whether this was also the case for their welfare impacts. We used $< 20,000 \text{ km}^2$ to define island size to remove large regions that could also be considered 'islands' (e.g., the UK). We produced separate contingency table tests for birds and ants in order to identify patterns specific to each of the two taxonomic groups. Tests comparing numbers of impacts using combined bird and ant datasets were not considered to be appropriate, as the bird dataset includes impacts caused by introduced species from an entire class (birds), whereas the ant dataset includes impacts caused by introduced species from one family (ants) of a class (insects).

Due to small numbers of impacts in some of the five impact severity categories, we merged them to form two categories: (i), (ii) and (iii) = 'less severe' impacts, (iv) and (v) = 'more severe' impacts. This intuitively separated impacts that are negligible, or that are already sustained by the affected animal ('less severe' impacts), from those that are more severe than previously experienced by the affected animal ('more severe' impacts). Each impact was then categorised as either being 'less severe' or 'more severe'. Dividing impacts in this manner is an approach adopted in previous studies using EICAT data[26,67]. Several categories of interest (in both the bird and ant datasets) had relatively few impacts—where feasible, these categories were pooled for the analysis[68] (for details on pooling, see the contingency table tests in Supplementary Note 3). We used exact tests for our analysis as the total sample size of each of our contingency table tests was below 1000[68], and because two tables contained cell values $< 5$[68].

### Reporting summary

Further information on research design is available in the Nature Portfolio Reporting Summary linked to this article.

### Data availability

The datasets generated in this study have been deposited in the Figshare database (https://doi.org/10.6084/m9.figshare.28553957.v2).

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

## Acknowledgements

The authors thank Caitlin Andrews, Kane Colston and Michaël Beaulieu who independently reviewed the AWICIS guidelines and assessment template and tested the framework by applying it to impact examples (as described in the Supplementary Information). Their feedback helped to clarify aspects of the guidelines and the instructions is the AWICIS assessment template. The authors thank an anonymous reviewer at Wild Animal Initiative (WAI) for commenting on a funding proposal describing this research, and four anonymous reviewers for their comments on a funding proposal describing this research submitted to the German Research Foundation (DFG). The authors thank Helen Roy at the UK Centre for Ecology and Hydrology (CEH) and Jonathan Jeschke at the Leibniz Institute for Freshwater Ecology and Inland Fisheries (IGB) for commenting on a detailed funding proposal describing this research. Funding: The authors gratefully acknowledge financial support from Wild Animal Initiative (WAI) (AH24-001, T.E.), German Research Foundation (DFG) (EV 304/4-1, T.E.) and BBSRC (BB/X014673/1, M.M.).

## Author contributions

T.E. conceived the framework, developed its scoring system and undertook the framework consultation; T.E. undertook the data collection, analyses and visualisations; M.M. oversaw the methods for the literature review; T.E. led the writing of the manuscript and completed the revisions; M.M. helped write the manuscript, and provided input on animal welfare concepts and assessment.

## Funding

## Competing interests

The authors declare no competing interests.
