## [Transparent Peer Review file · Nature Communications]

Quantifying and categorising the animal welfare impacts caused by biological invasions

Corresponding Author: Dr Thomas Evans

Version 0:

Reviewer comments:

Reviewer #1

(Remarks to the Author)

This is an interesting paper that investigates the individual welfare impacts of biological invasions on 'sentient' species. The authors have approached this problem by adopting existing frameworks and established datasets. While animal welfare is not (as) well-studied in biological invasions the cumulative impacts on populations and species leading to biodiversity loss (or compromise) are.

I believe that the authors have over-emphasised the novelty of their empirical work given that so much of it relies on previous frameworks, and datasets. The Impacts listed in Table 1 are in fact very well-studied in biological invasions (Competition, Predation, Hybridisation ...) and the novelty of their application to an individual's welfare – rather than a population impact – is not entirely clear to me, if the cumulative outcome is the same. I do agree that variability across individuals affecting population outcomes is interesting, and the particular impact on welfare is worth observation.

Is there a reason that 'Reproductive Fitness' is not considered as one of the Domain impacts (Lines 107 – 108)?

Given the low/poor reporting of welfare outcomes in studies, and the high-level (secondary datasets) used to assess the Framework I am concerned about the subjective repeatability of how severity is measured. As a Framework for future adoption I would like to see more consideration of how best-practice research could measure these impacts in future studies. It would have also been interesting to conduct an 'elicitation' process to test the repeatability of assigning measurements (and confidences) across a range of experts. This would provide substantially greater support for the outcomes/findings of the case-studies.

A lot of the language implies that the authors 'developed' and/or 'created' the datasets underpinning this paper, which I find to be a little disingenuous given that most of what they have done is adopt existing Frameworks and Datasets.

Figure 1a: Rather than list the legend alphabetically consider listing in the order presented on the X-axis

Table 3: The heading is uninformative. Table and Figure headings should be able to stand alone from the main text.

I am not enamoured by the large number of contingency table tests, or the small size of the cell counts in many cases. I suspect that a lot of basic assumptions are being violated across these tests.

Reviewer #2

(Remarks to the Author)

Reviewer #3

(Remarks to the Author)

This manuscript introduces AWICIS, a framework for classifying the animal welfare impacts of biological invasions, and applies it to global datasets of invasive birds and ants. The premise is innovative, and the application of welfare science to invasion ecology is an original direction. The figures are excellent, and the writing is generally clear.

However, the paper suffers from several core weaknesses that limit its scientific contribution in its current form. These include:

- an insufficiently rigorous and scientifically grounded definition of animal welfare and sentience;
- speculative interpretations based on weak empirical foundations;
- methodological over-reliance on secondary biodiversity data;
- and a tendency toward ideologically loaded framing, especially in its treatment of suffering and emotional states.

To conclude: This manuscript explores an important and novel direction by examining animal welfare impacts of biological invasions, but it is undermined by significant conceptual and methodological issues—most notably, the conflation of welfare, sentience, and indicators without clear definitions, speculative impact scoring from literature not designed for welfare assessment, and unsupported policy-oriented conclusions. To be suitable for a high-impact journal like *Nature Communications*, it would require a conceptual reframing to distinguish welfare from fitness, a more restrained and evidence-based tone, improved methodological justification, and a clearly structured, data-driven discussion. As it stands, the manuscript may be more appropriate for a specialist journal focused on invasion biology, animal ethics, or interdisciplinary conservation science.

Below is a detailed section-by-section review.

Title and Abstract

The term “alien species” is contentious. It carries negative connotations and is increasingly being replaced by “non-native” or “introduced” in scholarly work. See Soto et al. (2024) for a recent critique of the term’s sociopolitical baggage. Its use here undermines the neutrality expected in scientific communication.

The abstract includes suffering of non-native species and failed introductions as focal points, which stretches the framework’s conservation relevance. If the goal is to inform management or biosecurity, it is unclear why assessing the suffering of introduced animals that fail to establish is meaningful or actionable.

Introduction

The claim that “animals are sentient” is presented as an empirical fact applying broadly to all vertebrates and some invertebrates. This is an oversimplification of an ongoing scientific and philosophical debate. For example, the cited reference [5] (NY Declaration on Animal Consciousness) is a normative declaration, not empirical evidence. Many invertebrates and even some vertebrates still sit in a gray area regarding sentience.

The attempt to equate welfare with sentience-based suffering limits the framework’s applicability and risks conflating observable welfare indicators (injury, starvation, abnormal behavior) with inferred internal states (fear, anxiety, distress). While suffering is important, projecting emotional states onto wild animals, especially from indirect indicators, is a controversial and weakly supported practice.

Paragraphs describing interdisciplinary collaboration and public support are vague and somewhat disconnected from the core argument. The mention of biosecurity and public opinion seems like an attempt to make the study seem more policy-relevant than it actually is.

The paper does not provide a clear rationale for why welfare impacts should be studied separately from established biodiversity impacts. If welfare outcomes map directly onto fitness-reducing outcomes (injury, illness, death), then what is added by invoking welfare as a separate axis?

Methods

The impact mechanisms listed in Table 1 are not always mechanisms per se. For example, “structural changes to ecosystem” and “chemical changes to ecosystem” are environmental conditions that may enable welfare impacts, but are not direct mechanisms in the sense that predation or parasitism are. The list blurs causal chains, weakening analytical clarity.

The framework does not define clear, objective criteria for distinguishing severity levels. Terms like “short-term suffering” or “novel suffering” are not operationalized in a way that different assessors could reliably apply. The approach risks being subjective and inconsistent.

The reliance on literature-based biodiversity impact data is problematic. Most invasion-related literature is biased toward well-studied taxa, dramatic impacts, and high-income countries. The authors acknowledge this but continue to draw conclusions that generalize beyond the available data.

The treatment of impacts on introduced animals (e.g., two non-native species competing) adds conceptual confusion. What is the ecological or ethical relevance of these interactions within the context of a conservation or management-focused framework?

The absence of physiological indicators in most cases is treated as a data availability issue, but it may in fact reflect a deeper problem: that many “welfare impacts” inferred here are not readily observable or measurable, casting doubt on whether this framework captures real, meaningful animal experiences.

Results

The figures (especially Figures 1 and 3) are excellent, though some formatting overlaps (e.g., Figure 1c) should be corrected. The visualization of mechanisms, domains, and impact severity is helpful.

The method of counting “impacts” based on published studies raises several concerns:

- 1) It treats each publication as equally reliable, regardless of methodology.
- 2) It assumes geographic and taxonomic coverage is unbiased, which is demonstrably false.
- 3) It may conflate absence of evidence with evidence of absence, particularly in low-income regions or underrepresented taxa.
- 4) The decision to merge severity levels (i–iii vs. iv–v) is pragmatic but analytically limiting. It obscures variation within lower-level impacts, which may still be biologically or ethically relevant.
- 5) The high frequency of behavioral indicators, relative to physiological or direct observational data, suggests a potential over-reliance on interpretation. Many reported behaviors (e.g., nest abandonment, listlessness) could have multiple causes, not all linked to invasive species.

Discussion

The discussion is too long and poorly structured, mixing findings and speculation without a clear through-line. A more structured format (e.g., “Findings for Birds”, “Findings for Ants”, “Conceptual Limitations”, “Policy Implications”) would enhance readability and focus.

Some language is excessively rhetorical or anthropomorphic. Phrases like “death is both extremely painful and prolonged” or “panic behavior” need to be grounded in evidence, not intuition. Avoid emotionally loaded terminology unless backed by physiological or experimental data.

The repeated emphasis on islands as hotspots of welfare impacts mimics known biodiversity patterns, but the authors don’t explain why welfare should behave similarly. If predation and competition are more intense on islands, say so—but don’t assume a causal link without justification.

The assertion that AWICIS can inform management decisions or conservation planning is premature. The data are too uneven, the categories too subjective, and the framework too new to justify policy relevance at this stage.

The final paragraphs drift into speculative ethical territory. The suggestion that suffering during failed introductions should be considered when crafting animal welfare law may be ideologically driven rather than scientifically justified.

Reviewer #4

(Remarks to the Author)

Overview

The authors present a framework (AWICIS) for quantifying the impacts of biological invasions on animal welfare. This is an important problem which has not, to my knowledge, been covered by previous frameworks of invasive species impact assessment - for this reason, I think that the manuscript is a significant contribution to the field.

To categorise the welfare impacts of biological invasions, AWICIS adapts parts of existing frameworks for assessing the biodiversity impacts of invasive species (EICAT) and for animal welfare assessment (the Five Domains model), but is novel in the way that it combines these frameworks. In my view, the correspondence between AWICIS and EICAT is a strength of the manuscript, as it could facilitate the simultaneous application of both frameworks to a given biological invasion.

The next part of the framework, which quantifies impact severity by comparing impacts caused by invasive species with those caused by native species of the same class through the same mechanism (and also considers impact duration) is novel, practical to implement, and is a sensible comparison to make.

The authors also present a case study in which they apply their framework to welfare impacts of invasive birds and ants, based on analysing existing literature.

Overall, the authors present a highly detailed and thorough manuscript. I have some comments on both the framework itself, and on the case study, which I outline below.

Comments on the AWICIS framework

1. Evidencing the whole causal chain linking invasions to welfare. Providing evidence to demonstrate the welfare impacts of invasive species is the 4th stage of the AWICIS framework (Fig. S1), and the authors provide numerous examples in the supplementary tables and in the main text. However, what I think is missing in places is an explicit consideration of the entire causal chain which links the biological invasion to the welfare impact. For example, in Table S15 the authors present an example in which cheatgrass impacts the welfare of wapiti by increasing the intensity of wildfires. In this example, the

indicator is given as “Physical. Animals injured and killed by wildfires”. However, the evidence required to assign an impact severity of (v) is actually that cheatgrass increases the intensity of wildfires (otherwise, the impact could be graded (iii) if wildfires were of a similar intensity in the absence of cheatgrass). In other words, the causal chain is: cheatgrass → wildfires → welfare of wapiti, but only the latter link is evidenced. Therefore, I think it would be a good idea for the authors to discuss the importance of evidencing the full causal chain (especially in the section starting line 258 of the supplement), and to consider adding this requirement to the framework (Fig. S1).

2. Consideration of impact prevalence / extent. In my view, it is not a good idea to exclude changes to impact frequency from the consideration of “prevalence” (as stated in Table S1). For example, a native prey animal may have native predators, but predation by an invasive predator may be far more prevalent (due to e.g., native prey naïveté, or higher abundance of invasive predators than native predators). Consequently, a greater number of native prey animals may experience welfare impacts than they would if the invasive predator was absent. However, from my understanding of the framework, if the invasive predator does not increase the spatial extent of predation, then the impact would only be assigned severity ii or iii, regardless of the number of additional native animals which are preyed. Conversely, if the invasive predator increases the spatial extent of predation, then the impact could be assigned to category iv or v, even if the predation rate is low – to me, this seems inconsistent. A similar argument could be made about some of the other impact mechanisms, such as disease – would additional spread by the invasive species not impact the welfare of native individuals which would otherwise be unaffected? In my view, it would be better to allow for “increased prevalence” to include cases where there is clear evidence that biological invasion results in more frequent impacts, and for these cases to be assigned to the higher impact categories.

3. Examples of different impact mechanisms (Table 1 and Figure S2). It is not very clear how some of these mechanisms are defined (e.g., what is the difference between a “physical change” and a “structural change”, or between a “chemical change” and “poisoning/toxicity”?). I think it would be good to expand on these definitions (at least in the table if there isn’t room in the figure). Also, for some of the impact mechanisms, the authors did not find any examples of the mechanism in the literature. I have a couple of suggestions for the authors to consider:

- Physical changes to ecosystem: Carter et al. (2014). Exotic invasive plants alter thermal regimes: implications for management using a case study of a native ectotherm. *Functional Ecology*. <https://doi.org/10.1111/1365-2435.12374>

- Chemical changes to ecosystem: Hickman & Watling (2014). Leachates from an invasive shrub causes risk-prone behavior in a larval amphibian. *Behavioral Ecology*. <https://doi.org/10.1093/beheco/art121>

Comments on the case study

1. Statistical analyses. In their case study, the authors use contingency table tests to examine how the severity of welfare impacts (grouped into “more severe” and “less severe” impacts, which I think is fine) varies according to impact mechanism, location etc. However, there are some issues with the way these are reported. First, it is unclear which test was actually used – the table legends in Appendix B state that Fisher’s exact test was used, but the table reports χ^2 statistics, which Fisher’s exact test doesn’t use. Second, both Fisher’s exact test and the χ^2 test assume that the observations are independent, but this assumption could conceivably be violated (e.g., if one invasion generates impacts across multiple mechanisms, across multiple domains of animal welfare, or across multiple native species). Therefore, the authors should consider the assumptions of their statistical tests, and justify them in the methods section. Third, I could not find information how the tests were actually implemented (e.g., was a specific software package used, or were the calculations done by hand?). This detail should be provided in the methods section. Finally, the “statistics” section of the reporting summary checklist seems to contain some items where “n/a” was ticked when the authors could have ticked “confirmed” (e.g., the authors provide the full output from the contingency table tests in the supplementary tables, so could tick the item beginning “for null hypothesis testing...”).

Other comments

- The justification for applying the framework to birds and ants – that they are dissimilar taxa – is reasonable. However, readers may appreciate some brief context (either in the intro or the methods) to illustrate the importance of bird/ant invasions worldwide (e.g., what proportion of documented invasive species are birds/ants?).

- Lines 204-295 – to me this sentence implies that ants have a broader range of impacts despite the birds being more taxonomically diverse, when in fact there are at least several thousand more species of ants than birds – consider rephrasing?

- Lines 596 and 599 – shouldn’t islands be defined as “< 20,000 km²”, rather than “> 20,000 km²” (i.e., large ‘islands’ like the UK do not count)?

- Line 773 – the link to the reference doesn’t work

- Supplement lines 325-326 – I think the probabilities assigned to the low/medium/high confidence categories are unnecessary, and it is unclear how these probabilities could ever be objectively determined. In my opinion it is better just to have the low/medium/high labels, with the written explanation as the framework already requires.

- Table S3 – “livestock salivating” is provided as an example of an indicator, but from this table alone it’s unclear why this would be indicative of welfare impacts. Table S15 clarifies that the livestock exhibit excessive salivation, so the wording in

Table S3 should probably be changed to “livestock excessively salivating” or similar.

- Table S15 – in the South Africa / Marion Island example, the “impact mechanism” (5th column) refers to owls, when it should refer to mice.

Version 1:

Reviewer comments:

Reviewer #4

(Remarks to the Author)

I am happy to report that all of my comments from the first round of review have been addressed. I have no further comments to add on the revised version of the manuscript.

Reviewer #5

(Remarks to the Author)

Thanks for the opportunity to review this interesting manuscript. As I was asked to, I have checked the responses of the authors to the reviewers, and they seem to me to be exhaustive and convincing.

I only add a few comments here:

- One point that could be added is the sufferance of non-native individuals due to their management (e.g. trapping, killing – there are some data published on this). Although unavoidable, this is surely something that occurs. In this regard, cautious writing is also needed to avoid biological invasions negationists and animalists using this argument to stop management efforts; I would stress the need to minimize the number of suffering animals via early and rapid management, so that “only” a few animals will suffer, compared to what happens over time when interventions are late and populations became too large.

-Checking the more relevant literature, I found these two papers that I think should be included in the manuscript, given their very pertinent topic:

- Carneiro, L., Leroy, B., Capinha, C., Bradshaw, C. J., Bertolino, S., Catford, J. A., ... & Courchamp, F. (2025). Typology of the ecological impacts of biological invasions. *Trends in Ecology & Evolution*, 40(6), 563-574.
- Haubrock, P. J., Everts, T., Abreo, N. A. S., Bojko, J., Deklerck, V., Dickey, J. W., ... & Britton, J. R. (2025). The impacts of biological invasions. *Biological Reviews*.

Response to comments

We thank all four reviewers for their comments, to which we have responded in blue text. We have highlighted changes to the revised manuscript in yellow, and provided line number references in our responses below.

Reviewer #1

(1) This is an interesting paper that investigates the individual welfare impacts of biological invasions on ‘sentient’ species. The authors have approached this problem by adopting existing frameworks and established datasets. While animal welfare is not (as) well-studied in biological invasions the cumulative impacts on populations and species leading to biodiversity loss (or compromise) are.

I believe that the authors have over-emphasised the novelty of their empirical work given that so much of it relies on previous frameworks, and datasets. The Impacts listed in Table 1 are in fact very well-studied in biological invasions (Competition, Predation, Hybridisation ...) and the novelty of their application to an individual’s welfare – rather than a population impact – is not entirely clear to me, if the cumulative outcome is the same. I do agree that variability across individuals affecting population outcomes is interesting, and the particular impact on welfare is worth observation.

(1.1) Response: The purpose of the AWICIS framework is to identify and assess the severity of impacts that harm the welfare of individual animals – not population level (biodiversity) impacts that affect the survival of entire species (their ability to reproduce and maintain viable populations). No such animal welfare framework has been developed or published. It is true that biodiversity impact mechanisms (competition, predation etc.) are well studied – but we know little about how severely these mechanisms affect the welfare of individual animals. AWICIS has been developed to improve our knowledge in this respect.

(1.2) Response: The results of our assessment for birds and ants suggest that the cumulative outcome of population level (biodiversity) impacts and the animal welfare impacts caused by biological invasions are not the same. For example, for birds, an entire impact mechanism (hybridisation) has been reported to have severe biodiversity impacts in past published assessments (for example, hybridisation between introduced mallards (*Anas platyrhynchos*) threatens the survival of native dabbling duck species (*Anas* spp.) (<https://doi.org/10.1071/PC140041>). However, using AWICIS our assessment found

hybridisation with introduced birds to only cause level (i) (negligible) animal welfare impacts. Other mechanisms that cause severe animal suffering may also not have severe biodiversity impacts – for example, parasites may not often cause species extinctions (because as a species' population declines as a result of parasitism, transmission of the parasite is reduced to low densities (<https://www.nature.com/scitable/knowledge/library/ecological-consequences-of-parasitism-13255694/>). However, parasites can cause severe suffering of individual host animals, as we show with the example of the avian vampire fly (*Philornis downsi*) parasitising native birds on the Galápagos Islands (<https://doi.org/10.1038/s41598-021-94996-7>) – this welfare impact is assessed as level (v) using AWICIS.

(1.3) Response: We acknowledge that AWICIS does adopt the biodiversity impact mechanisms (e.g., competition, predation) that have been developed for the EICAT framework, which assesses impacts on biodiversity caused by introduced species. We adopted these biodiversity impact mechanisms because it makes sense to do so, as they are the mechanisms through which biological invasions cause welfare impacts on individual animals. It's worth noting here that Reviewer 4 supports the approach we have adopted (see Reviewer 4 comments – points 34, 35 and 36 below). We should have been clearer about why we adopted the EICAT mechanisms in the introduction – in response to this comment, we have added a section to clarify (lines 129 – 138 in the main manuscript).

(1.4) Response: AWICIS contains novel approaches that have not been used for other invasion biology frameworks – particularly in terms of the way severity is quantified for animal welfare impacts. Novel aspects of the framework include:

- its scope, which enables the assessment of welfare impacts on native and introduced species (and also wild and domesticated species) – EICAT and other frameworks developed to assess impacts on non-human organisms caused by biological invasions only consider impacts on native species.
- the five impact severity categories (i – v) which enable relative changes to the welfare of an individual animal caused by biological invasions to be identified.
- the indicator focussed approach – AWICIS requires the use of indicators (physical, behavioural and physiological indicators) to provide evidence of animal welfare impacts.
- the development of different criteria that must be applied when assessing welfare impacts on native vs introduced species.
- definitions for 'short-term' and 'prolonged' animal welfare impacts which are used to distinguish between the five different impact severity categories.

(1.5) Response: In response to the comment by the reviewer, we have added additional text to the introduction to state why we should study welfare impacts, and why the new framework is needed (lines 86 – 105 – inserted below for reference). We acknowledge that we didn't provide enough information on this in the previously submitted manuscript.

“This framework is needed because the animal welfare impacts of biological invasions are a different type of impact to the biodiversity impacts of biological invasions (as assessed using the EICAT framework)(21, 26) – animal welfare impacts affect the mental and physical state of individual animals(4); biodiversity impacts affect the survival of entire species (their ability to reproduce and maintain viable populations). Furthermore, although the mechanisms through which biological invasions cause animal welfare and biodiversity impacts may often be the same (e.g., predation of a native species by an introduced species, which can cause declines in the population of the native species or even its extinction(27), and also the welfare of an individual of that species, which may suffer physically and emotionally whilst being preyed on(28)), the severity of the animal welfare and biodiversity impacts caused through the same mechanism may not be congruent. For example, introduced parasites may not cause species extinctions (a severe biodiversity impact), because as a species' population declines as a result of parasitism, transmission of the parasite is reduced to low densities(29) – however, introduced parasites can cause severe suffering of individual host animals(15). Therefore, AWICIS may provide information on the severity of animal welfare impacts caused by biological invasions that cannot be obtained using existing frameworks such as EICAT. Finally, a dedicated animal welfare framework is required because biological invasions can affect the welfare of both native and introduced species, but existing frameworks that assess the impacts of biological invasions on non-human organisms (including EICAT(21, 26)) have been designed solely to assess impacts on native species (but not introduced species).”

(1.6) Response: In summary, the two main aims of invasion biology frameworks are to (i) quantify by severity and (ii) categorise by type the impacts of introduced species. EICAT, and AWICIS both do this. AWICAT's impact severity quantification methods are novel; AWICAT's categorisation methods adopt those used for EICAT (i.e., the EICAT biodiversity impact mechanisms) because it is appropriate to do so. Using these EICAT biodiversity mechanisms to categorise animal welfare impacts is novel. As the severity of the animal welfare and biodiversity impacts of biological invasions caused through the same

mechanisms are not congruent, a separate framework (AWICIS) is required to quantify welfare impacts. Further, AWICIS is needed because EICAT only assesses impacts on native species, but animal welfare impacts can affect native and introduced species.

(1.7) Response: We did acknowledge in the original submitted manuscript that the mechanisms (competition, predation etc.) used in AWICIS were adopted (and amended) from EICAT – we included a paragraph to describe how some of these mechanisms had been amended to make them more suitable to assess welfare impacts. However, we have checked the revised manuscript and added additional text to make sure this is clear:

- we added a footnote to Table 1 in the main manuscript which describes the mechanisms (lines 141 – 142):

“The impact mechanisms described in this table are adopted (and amended) from a published framework developed to assess the impacts of introduced species on native biodiversity – the Environmental Impact Classification for Alien Taxa (EICAT)(21, 26).”

- we added the following text to the legend for Figure S2 in the Supporting Information (lines 101 – 103):

“Figure S2. AWICIS – impact mechanisms. The impact mechanisms described in this figure are adopted (and amended) from a published framework developed to assess the environmental impacts of introduced species on native biodiversity – the Environmental Impact Classification for Alien Taxa (EICAT)(21).”

(1.8) Response: We have also added text to the overview figure in the Supporting Information (Fig. S1 at line 73) to make sure it is clear the mechanisms are adopted from EICAT (see Section 2 of the figure – ‘Categorising impacts’).

(2) Is there a reason that ‘Reproductive Fitness’ is not considered as one of the Domain impacts (Lines 107 – 108)?

(2.1) Response: The five domains are those described in a published model (<https://doi.org/10.3390/ani10101870>). We integrated the domains model with the AWICIS

framework to demonstrate how they are linked, and to show how animal welfare impacts quantified and categorised under AWICIS could be mapped across the domains.

(2.2) Response: We believe that reproductive fitness is a broad outcome that could be affected by impacts relevant to several of these domains (e.g., Domain 1: Nutrition – poor nutrition could affect reproductive health; Domain 3: Health – disease could affect reproductive health).

(3) Given the low/poor reporting of welfare outcomes in studies, and the high-level (secondary datasets) used to assess the Framework I am concerned about the subjective repeatability of how severity is measured. As a Framework for future adoption I would like to see more consideration of how best-practice research could measure these impacts in future studies. It would have also been interesting to conduct an ‘elicitation’ process to test the repeatability of assigning measurements (and confidences) across a range of experts. This would provide substantially greater support for the outcomes/findings of the case-studies.

(3.1) Response: We have produced an assessment template which we provide as best practice for reporting animal welfare impacts using AWICIS. Section 7 of the guidance document (lines 416 – 423 in the Supporting Information) provides a link to the assessment template, which has been uploaded to a repository (<https://figshare.com/s/29a0492309398e111c18>). T.E. produced the reporting template for EICAT and has used this experience when developing the AWICIS reporting template. It provides structure to the assessment process, ensuring all relevant information is considered and reported. The template includes several impact examples (these were the examples provided in Table 15 of the Supporting Information in the originally submitted manuscript, which we have deleted).

(3.2) Response: EICAT was developed in 2013 and published in 2014 (<https://doi.org/10.1371/journal.pbio.1001850>). It was eventually adopted by the IUCN in 2020 (<https://doi.org/10.2305/IUCN.CH.2020.05.en>). EICAT was the subject of several workshops prior to its formal adoption by the IUCN, and has been comprehensively tested and revised since its publication in 2014. For example, these revisions include changes to some of the mechanisms (<https://doi.org/10.3897/neobiota.62.52723>). If AWICIS is published, we expect that it would be tested and revised over many years in a similar manner to EICAT (even if not formally adopted by the IUCN or another organisation). However, to assess its applicability, we have undertaken a test exercise which is described in Appendix B of the Supporting Information (lines 425 – 527). The methods and results are

also briefly described in the revised manuscript (lines 196 – 205 and 219 – 229). In brief, having made all the revisions to the framework as suggested by the reviewers, we asked three research scientists to:

- read the guidelines and the assessment template and provide feedback (this in itself was useful as it resulted in additional (minor) clarifications to the guidelines and the template).
- undertake training by completing the template for five 'training' impact examples which were reviewed by T.E. who provided the assessors with feedback for corrections.
- complete the template for a further ten impact examples (without assistance).

(3.3) Response: We emphasise here that there are no 'experts' with prior knowledge of the framework – so the assessors did have quite a lot of new information to digest.

Nevertheless, the results indicate that AWICIS can be used by researchers that are not familiar with the framework to assess animal welfare impacts. T.E. reviewed the assessments and provided the assessors with feedback. The reviewers agreed with all review comments (i.e., following review, there were no major differences of opinion regarding the assessment process and outcomes). There were some errors with some of the assessments, but this is to be expected given that the assessors were unfamiliar with the framework. These errors were not attributed to some fundamental flaw with the framework. Indeed, formal EICAT assessments that are submitted to the EICAT Authority are always reviewed and often sent back to the assessors for corrections. T.E. reviews EICAT assessments submitted to the EICAT Authority, and many require corrections. Examples of errors made by the assessors include:

- assessment of the severity of a welfare impact on Roe deer (*Capreolus capreolus*) caused by competition with Crested porcupines (*Hystrix cris*) as being level (ii), when actually the impact was more severe – after review by T.E., the assessor acknowledged that they had assessed the impact of competition between the two animals at feeding stations, but had forgotten to consider the effect of injuries sustained by deer that were caused by porcupine quills.
- assessment of an impact as being short-term, but then incorrect allocation of the impact to a severity category associated with prolonged impacts (level v) – when T.E. commented that the impact severity category needed checking, the assessor allocated it to the appropriate impact category (level iv).

- categorisation of an impact caused by a parasitic flea as being ‘Disease transmission’ when it could more accurately have been categorised as ‘Parasitism’.

(3.4) Response: Indirect impacts are more complex to assess than direct impacts. Two assessors incorrectly assessed an indirect impact example – however after discussions with one of the assessors it became clear that this was because of the way T.E. had structured the example in the template. We have provided text on this in the Supporting Information (lines 492 – 507, reproduced below for reference). Importantly though, the two assessors did correctly assess the other indirect impact example in the assessment template (‘chemical impacts to ecosystems’ – the fourth of the ten impact examples in the template at Row 8). This indicates that indirect impacts can be assessed using the framework – they are just a little more complex to understand, and the assessors were unfamiliar with the process.

*“Two assessors incorrectly categorised an ‘Indirect impact through interactions with other species’ as being ‘Predation’ (Assessment templates 1 and 3 – Row 12, Column Z). After discussion with one of the assessors, it became apparent that the way the impact example had been structured in the template by T.E. had caused some confusion. The impact was for increased predation of Macquarie Island parakeets (*Cyanoramphus novaezelandiae erythrotis*) on Macquarie Island – this increase was caused by the introduction of European rabbits (*Oryctolagus Cuniculus*), which provided food for feral cats (*Felis catus*), increasing the cat population on the island and hence the prevalence of cat predation of individual birds. T.E. had included the feral cat in the template as the first interacting species, and the European rabbit as the second interacting species, when actually the European rabbit should have been included as the first interacting species (as this is the species causing the indirect impact) and the feral cat should have been included as the second interacting species (as increased predation by feral cats was the outcome – the secondary mechanism). We have updated the instructions in the assessment template to provide advice on the order in which interacting species should be listed for the mechanism ‘Impacts caused by interactions with other species’ (see Row 1, Columns L – O of the assessment template). The assessors confirmed that this would improve the assessment process for this mechanism, by making it more intuitive.”*

(3.5) Response: The three assessors were selected because they have different levels of experience and areas of expertise, and because they had no prior knowledge of AWICIS. Caitlin Andrews is a conservation scientist with experience in the use of frameworks, having

used EICAT to undertake risk assessments associated with the introduction of a threatened bird species to a new island habitat (<https://doi.org/10.1007/s10530-024-03341-2>); Michaël Beaulieu is an animal welfare researcher with a good understanding of the ways in which indicators can be used to assess animal welfare (<https://doi.org/10.1111/brv.13009>); Kane Colston is an animal welfare researcher (PhD student) and a graduate of the MSc in Global Wildlife Health and Conservation at the University of Bristol (<https://www.bristol.ac.uk/study/postgraduate/taught/msc-global-wildlife-health-and-conservation/>).

(3.6) Response: We agree that providing advice on how best-practice research could measure welfare impacts in future studies is important. AWICIS is an indicator focussed framework – we have added text to the discussion which may help to direct future welfare research efforts, stating that it should focus on gathering evidence in the form of the three types of indicator described in the AWICIS guidelines (physical, behavioural and physiological indicators). We emphasise that future biodiversity research (that is not concerned with welfare impacts) could still provide useful observations – hence we also encourage biodiversity researchers to take advantage of opportunities to gather data relevant to welfare impacts (lines 498 – 505 in the main manuscript). We note that biodiversity research does not often consider welfare impacts on individual introduced animals (understandably, it is concerned with the survival of species) – we suggest that future biodiversity research could include welfare assessments using AWICIS for both native *and* introduced animals (lines 505 – 513 in the main manuscript). We have also stated that data is scarce for the third type of indicator (physiological data), and that welfare studies of wild animals would benefit from more of this research, which appears to be neglected – we discuss the reasons why this may be the case (lines 457 – 474 in the main manuscript).

(4) A lot of the language implies that the authors ‘developed’ and/or ‘created’ the datasets underpinning this paper, which I find to be a little disingenuous given that most of what they have done is adopt existing Frameworks and Datasets.

(4.1) Response: We do appreciate the reviewer’s comments which are useful, but we think it’s important to state here that we did not adopt existing datasets. What we did was review biodiversity research articles on the impacts of biological invasions, and assess the welfare impacts described in those articles following the AWICIS guidelines. In many cases, the information was adequate to confidently assess welfare impact severity because the mechanisms that cause biodiversity impacts on species also cause animal welfare impacts

on individuals. For example, almost all of our ant invasion data was associated with predation – the descriptions of predation were sufficient to assess welfare impacts.

(4.2) Response: The process we adopted (reviewing articles and assessing impacts using AWICIS framework guidelines) is the same process used to undertake EICAT assessments (research articles and other information on biological invasions are reviewed to assess biodiversity impacts, following EICAT guidelines). Indeed, EICAT assessments may include information that has not necessarily been published to assess biodiversity impacts – for example, this information may simply be observations of interactions between two species, and not primary literature on biodiversity impacts. Hence, we believe that the welfare impact severity datasets (impacts from i to v) produced using AWICIS are novel (no less so than those produced using EICAT). We have reviewed the methods section to make sure it is clear that we did not adopt any existing datasets for our study – but we do clearly explain that we reviewed biodiversity research articles to assess their welfare impacts, and we discuss the limitations associated with this in the discussion (lines 371 – 383 in the main manuscript).

(4.3) Response: We used the term ‘created’ in the introduction when referring to the animal welfare impact severity datasets that have been produced by applying the AWICIS framework to published biodiversity research articles of biological invasions. These are global AWICIS datasets for bird and ant invasions with animal welfare impact severity scores of (i) – (v). We do think that these animal welfare impact severity datasets are novel. Nevertheless, we have taken onboard the reviewer’s comment – we have amended the introduction to remove use of the term ‘created’ (lines 209 – 213 in the main manuscript). It now reads as follows:

“To test the framework’s utility in assessing different types of welfare impacts caused by widely separated taxa, we applied it to biological invasions associated with two groups of introduced species – birds and ants. In so doing, we classified the animal welfare impacts of biological invasions by their severity and type, and we used these data to investigate geographic and taxonomic patterns in their distribution.”

(4.4) Response: We used the term ‘developed’ (e.g., “the AWICIS framework developed for this study”, and “we developed a framework”) because we do think that we have developed a novel framework to assess animal welfare impacts of biological invasions. Our framework includes a novel scoring system that is different to scoring systems adopted in other frameworks such as EICAT; it considers impacts on native and introduced, and wild and

domesticated species, which is a novel approach within the field of invasion biology; it uses established biodiversity impact mechanisms to assess welfare impacts because it is appropriate to do so (and in itself, this is novel); development of the framework required an interdisciplinary collaboration between the two authors. Nevertheless, we take the reviewer's point, and we do not want to appear disingenuous. We have reviewed the manuscript, removing the term 'developed' when referring to the framework. In the abstract we have replaced 'developed' with 'present' (line 15):

"We ~~developed~~ present a framework which can be used to identify relative changes to the welfare of an individual animal caused by biological invasions."

(4.5) Response: In the introduction we have amended the 6th paragraph as follows (lines 85-86):

"Here, we test a new framework which can be used to assess the animal welfare impacts caused by biological invasions"

(5) Figure 1a: Rather than list the legend alphabetically consider listing in the order presented on the X-axis

(5.1) Response: Thank you – we have updated the legend order as suggested, and the figure is now easier to interpret. We have done the same for Figure 3 as well.

(6) Table 3: The heading is uninformative.

(6.1) Response: We have changed the title of the table. We have moved the additional text to below the table.

(7) Table and Figure headings should be able to stand alone from the main text.

(7.1) Response: We have updated all figures to provide a brief title sentence. We have followed this with a short statement of what is depicted (as stipulated in the journal's author guidelines).

(7.2) Response: With the amendments we have made to Table 3, all the tables in the manuscript meet the requirements stipulated by the reviewer (and the author guidelines).

(8) I am not enamoured by the large number of contingency table tests, or the small size of the cell counts in many cases. I suspect that a lot of basic assumptions are being violated across these tests.

(8.1) Response: We have critically reviewed our approach to the contingency table tests. There were 11 in the submitted manuscript – on reflection, five of these tests were not particularly useful, so we have deleted them. The deleted tests were as follows:

- two tests analysing the distribution of impact severity across continents (one test for ants and one test for birds). We do not think examining patterns at this broad spatial scale provides useful insights other than to reinforce patterns that we have already shown using the global map of impacts.
- two tests describing the distribution of impact severity across domains (one test for ants and one test for birds). We have already tested impact severity across impact mechanisms – the links between the domains and the mechanisms are shown in Fig. 1 and Fig. 3. In short, impact associated with the mechanism ‘predation’ tended to be more severe for both bird and ant invasions – Fig. 1 and Fig. 3 clearly display which domains are associated with predation.
- one test examining the distribution of impact severity across status of affected animal (either native or introduced) for bird invasions. There were no ‘more severe’ impacts reported for introduced animals affected by these invasions (see table below). Therefore, we think it is more appropriate (and straightforward) to just describe this result (lines 280 – 282 in the main manuscript).

	Introduced	Native
Less severe	305	472
More severe	0	79

- Note: we did not test impact severity across status of affected animal (native or introduced) for ant invasions in the original manuscript as there were no reports of wild, introduced animals being affected by ant invasions.

(8.2) Response: The manuscript now includes six contingency table tests in total (three each for bird and ant invasions, respectively). They are presented more concisely in the supporting information (three tests to a single A4 portrait page) rather than spanning several landscape pages as in the originally submitted supporting information. As such it is easier to digest the information they contain.

(8.3) Response: Following the advice in the Handbook for Biological Statistics (<https://www.biostathandbook.com/small.html>) we pooled some categories to reduce the number of cells with small values (see 'Pooling' section in <https://www.biostathandbook.com/small.html>). We have provided details on the pooled categories below each contingency table test in the Supporting Information (Appendix B). Also following advice in the Handbook for Biological Statistics, we used exact tests, as all our tables had total sample sizes of less than 1000 (see 'Recommendation' section in <https://www.biostathandbook.com/small.html>). We also used exact tests because after pooling, three cells in tests for ant invasions did contain values < 5 (but none had 0 values), and 5 is considered a 'cut-off' for using exact tests (see 'The problem with small numbers' section in <https://www.biostathandbook.com/small.html>). We have described this in the main manuscript (lines 635 – 639). We undertook the analysis using the FunChisq package in R – we have included this information in the supporting information (at the start of Appendix C – line 531) and the main manuscript (line 609). The results of the revised contingency table analysis are similar to those produced by the original analysis – they do not change the broad conclusions presented in the discussion.

Reviewer #2

(9.1) Response: Thank you for your review.

Reviewer #3

(10) This manuscript introduces AWICIS, a framework for classifying the animal welfare impacts of biological invasions, and applies it to global datasets of invasive birds and ants. The premise is innovative, and the application of welfare science to invasion ecology is an original direction. The figures are excellent, and the writing is generally clear.

(10.1) Response: Thank you.

(11) However, the paper suffers from several core weaknesses that limit its scientific contribution in its current form. These include:

- an insufficiently rigorous and scientifically grounded definition of animal welfare and sentience;
- speculative interpretations based on weak empirical foundations;
- methodological over-reliance on secondary biodiversity data;
- and a tendency toward ideologically loaded framing, especially in its treatment of suffering and emotional states.

(11.1) Response: We have addressed the above four broad issues by responding to the specific comments provided by the reviewer below.

(12) To conclude: This manuscript explores an important and novel direction by examining animal welfare impacts of biological invasions, but it is undermined by significant conceptual and methodological issues—most notably, the conflation of welfare, sentience, and indicators without clear definitions, speculative impact scoring from literature not designed for welfare assessment, and unsupported policy-oriented conclusions. To be suitable for a high-impact journal like Nature Communications, it would require a conceptual reframing to distinguish welfare from fitness, a more restrained and evidence-based tone, improved methodological justification, and a clearly structured, data-driven discussion. As it stands, the manuscript may be more appropriate for a specialist journal focused on invasion biology, animal ethics, or interdisciplinary conservation science.

(12.1) Response: We have addressed these comments by responding to the specific comments provided by the reviewer below. In summary, some of the main changes we have made include:

- providing a definition of ‘animal welfare’ and a comparison with impacts affecting species survival (i.e., biodiversity impacts) – linking AWICIS (which assesses animal welfare) with a related framework (EICAT) which assesses species survival.
- removing the requirement for AWICIS to only assess welfare impacts on animals that are considered to be sentient (we acknowledge that this may cause some confusion and debate).
- improved methods, including revisions to the contingency table analysis and further information to justify the use of biodiversity impact data to score animal welfare impacts.
- restructuring the discussion, providing discrete sections with headings (we also added sections to the introduction) – we have been clear about the limitations of the study by placing the limitations section at the start of the discussion.

- editing the manuscript to remove vague text that is secondary to the purpose of AWICIS (such as its potential indirect benefits for biodiversity conservation, which we felt might confuse the reader), and to remove speculative text – as a result, the discussion is much more concise.
- editing the manuscript to remove emotional language – it now has a more restrained, evidence-based tone.

(13) Below is a detailed section-by-section review.

Title and Abstract

The term “alien species” is contentious. It carries negative connotations and is increasingly being replaced by “non-native” or “introduced” in scholarly work. See Soto et al. (2024) for a recent critique of the term’s sociopolitical baggage. Its use here undermines the neutrality expected in scientific communication.

(13.1) Response: We have reviewed the manuscript and supporting information to remove use of the term ‘alien’ species – we instead use the term ‘introduced’. We have also used the term ‘introduced’ in this document.

(13.2) Response: We used the term ‘alien species’ in the original manuscript to be consistent with other related frameworks which use the term alien species, e.g., the Environmental Impact Classification for Alien Taxa (EICAT) (<https://doi.org/10.1371/journal.pbio.1001850>) and the Socio-economic Impact Classification for Alien Taxa (SEICAT) (<https://doi.org/10.1111/2041-210X.12844>). However, we do understand the point the reviewer is making here. Fortunately, the title of our framework doesn’t include the term alien species. As it assesses welfare impacts that affect native and introduced animals, we felt it more appropriate to use the term ‘invasion science’ in the title instead of ‘alien taxa’ (hence, ‘Animal Welfare Impact Classification for Invasion Science, AWICIS’, rather than ‘Animal Welfare Impact Classification for Alien Taxa, AWICAT’).

(14) The abstract includes suffering of non-native species and failed introductions as focal points, which stretches the framework’s conservation relevance. If the goal is to inform management or biosecurity, it is unclear why assessing the suffering of introduced animals that fail to establish is meaningful or actionable.

(14.1) Response: The goal of the framework is not to inform conservation efforts (or to protect biodiversity in any way) – it is to protect the welfare of individual animals. We have amended the guidelines in the Supporting Information (lines 21 – 37) to make it clear that the

purpose of the framework is to assess welfare impacts. We have also amended **Fig. S1** at **line 73** in the Supporting Information (see Section 1. 'Data'). The revised text in the guidelines is included here for ease of reference.

"1.1 Defining animal welfare

AWICIS adopts the following broad definition of animal welfare, which is taken from the World Organisation for Animal Health (<https://www.woah.org/en/what-we-do/animal-health-and-welfare/animal-welfare/>):

'the physical and mental state of an animal in relation to the conditions in which it lives and dies'(4).

1.2 Animal welfare vs species survival

AWICIS assesses the impacts of biological invasions on the welfare of animals. Other frameworks have been published which assess different types of impacts that are caused by biological invasions. These frameworks include the Environmental Impact Classification for Alien Taxa (EICAT)(21, 26), which assesses the impacts of introduced species on native biodiversity. Hence, EICAT assesses impacts that affect the survival of native species, including their ability to reproduce and maintain viable populations. AWICIS does not assess impacts on species survival."

(14.2) Response: We have deleted from the introduction (in the main manuscript), the text describing the potential benefits of the framework for biodiversity conservation, as we think this may cause some confusion regarding the purpose of the framework. The deleted text is reproduced below.

~~*"Interdisciplinary research can lead to important advances in the understanding of biological invasions(14). Identifying and publicising the animal welfare impacts of biological invasions may forge collaborations between conservation science and animal welfare researchers that result in a more holistic understanding of the impacts of biological invasions on both biodiversity and animal welfare. Furthermore, although the biodiversity impacts of some alien species provide a compelling reason to develop rigorous biosecurity protocols to prevent their introduction, biosecurity remains poor in many regions(15). Identifying and publicising the biodiversity impacts of biological invasions justifies actions to improve biosecurity(16); identifying and publicising the*~~

~~animal welfare impacts of biological invasions may achieve the same outcome, particularly as there is broad public support for animal welfare(17).”~~

(14.3) Response: The purpose of the framework is to assess the animal welfare impacts of biological invasions. These welfare impacts can affect both native and introduced animals. As such, we believe that welfare impacts on introduced individual animals caused by biological invasions are relevant. So we have maintained the scope of the framework to include impacts on introduced animals. However, we agree with the reviewer that the text in the discussion on failed introductions was not directly relevant to the manuscript – we have deleted it. The deleted text is reproduced below.

~~“It could be concluded then, that biological invasions overwhelmingly affect native, but not alien birds. However, we know little about the welfare impacts sustained by alien animals that are introduced to a new location but that do not survive, or perhaps survive briefly enough to establish a small alien population which eventually dies out. The Global Avian Invasions Atlas (GAVIA)(48) has catalogued thousands of introductions of alien birds that have occurred over the last century—some of these introductions have been deliberate (e.g., jungle mynas (*Acridotheres fuscus*) released during spiritual ceremonies in Taiwan)(49), others unintentional (e.g., African sacred ibises (*Threskiornis aethiopicus*) which have escaped from zoos in France)(47). Establishment success for introduced alien birds is strongly influenced by location-level processes, and in particular, climate suitability(50). Hence, the animals that do not establish are often unsuited to the environments that they are introduced to, and are likely to suffer from welfare impacts such as exposure and starvation prior to their death. If, as research suggests, it is generalist animals that survive these introductions, then it is likely to be non-generalists (e.g., those with specific habitat requirements and hence restricted native ranges) that tend to die out(51). Due to their restricted native ranges, these non-generalist individuals are more likely to suffer from unique (and hence severe) welfare impacts as aliens when compared to the impacts they are affected by in their native ranges. As the introduction of alien species continues to rise, including for birds(52), it is likely that animals will continue to be introduced to unsuitable environments (sometimes deliberately), where their suffering goes unreported because they perish (i.e., do not establish alien populations that are studied). Hence, biological invasions are likely to affect the welfare of both native and alien animals, despite the story told by our results. In the UK, the Animal Welfare Act 2006 requires owners and keepers to ‘provide their animals with suitable environment, diet and the ability to exhibit normal behaviours.’ It includes provisions to cover the~~

~~deliberate release of native, domesticated animals(53)—extending this legislation to also protect alien animals may help to prevent their unnecessary, deliberate introduction, benefiting both their welfare and biodiversity.”~~

(14.4) Response: We also deleted the reference to failed introductions in the abstract of the main manuscript.

(15) Introduction

The claim that “animals are sentient” is presented as an empirical fact applying broadly to all vertebrates and some invertebrates. This is an oversimplification of an ongoing scientific and philosophical debate. For example, the cited reference [5] (NY Declaration on Animal Consciousness) is a normative declaration, not empirical evidence. Many invertebrates and even some vertebrates still sit in a gray area regarding sentience.

(15.1) Response: We have modified the text when introducing the issue of sentience (which we think is important to touch on in the introduction) and acknowledge that there is uncertainty about the prevalence of sentience across species (lines 40 – 49 of the main manuscript). We have reproduced the text here for reference:

“The scientific study of animal welfare has been developing since the 1960s (e.g., Dawkins, 1980(5); Fraser and Broom, 1996(6); Mendl et al., 2017(3); Appleby et al., 2018(7); Mason et al., in press(8)), and there is growing consensus that the ability to consciously experience emotions and sensations (sentience) is a key determinant of animal welfare(9). Whilst we cannot be certain whether and which non-human species are sentient, it is increasingly accepted, based on behavioural, cognitive and neuroscientific evidence, that mammals and birds, and likely other vertebrates and some invertebrates, have this capacity(10) – they are sentient beings(11) and treated as such in legislation(12). Hence, in threatening or harmful situations, animals are likely to experience suffering through negative affective (emotional) states(13), including discrete states such as anxiety, pain and fear(14), as well as potentially detrimental changes to their physical state.”

(16) The attempt to equate welfare with sentience-based suffering limits the framework’s applicability and risks conflating observable welfare indicators (injury, starvation, abnormal behaviour) with inferred internal states (fear, anxiety, distress). While suffering is important, projecting emotional states onto wild animals, especially from indirect indicators, is a

controversial and weakly supported practice.

(16.1) Response: Thank you for pointing this out – we agree that there is much to understand regarding sentience, and therefore using it to stipulate which animals can and cannot be assessed using AWICIS may cause confusion and debate. We have removed the text in the manuscript relating to the requirement to assess sentient animals.

(16.2) Response: We have included a definition of sentience, being *‘the ability to have physical and emotional experiences, which matter to the animal, and which can be positive and negative’*(30). We state that whilst AWICIS can be used to assess welfare impacts of biological invasions on both vertebrate and invertebrate species, when considering mental states to be the key determinant of welfare, inferences about such states can be made most strongly for animals that are protected under the *Animal Welfare (Sentience) Act 2022* (lines 121 – 127 of the main manuscript – text reproduced below for reference). The same text is also included in the Supporting Information (lines 41 – 47).

“Animal sentience is ‘the ability to have physical and emotional experiences, which matter to the animal, and which can be positive and negative’(30). AWICIS can be used to assess the impacts of biological invasions on the welfare (physical and mental state) of both vertebrate and invertebrate species. However, when considering mental states to be the key determinant of welfare, inferences about such states can be made most strongly for animals that are protected under the Animal Welfare (Sentience) Act 2022(12) and assumed, at least in UK legislation, to be sentient. Currently, these are non-human vertebrates, cephalopod molluscs and decapod crustaceans.”

(16.3) Response: Also in response to a previous comment by the reviewer (point 12 above), we have distinguished ‘welfare’ from ‘species survival’ (lines 30 – 37 in the Supporting Information), including in the overview figure (Fig. S1 at line 73 in the Supporting Information (see Section 1. ‘Data’)). The text from the guidelines is reproduced in point 14.1 above.

(17) Paragraphs describing interdisciplinary collaboration and public support are vague and somewhat disconnected from the core argument. The mention of biosecurity and public opinion seems like an attempt to make the study seem more policy-relevant than it actually is.

(17.1) Response: We agree that the introduction should focus solely on describing the key purpose of the framework, so we have deleted this text. Indeed, we think this text could possibly confuse readers as it suggests the framework has been developed to encourage actions to protect biodiversity. This isn't the case, although it may be a beneficial indirect outcome of the framework. The deleted text is reproduced below.

~~*“Interdisciplinary research can lead to important advances in the understanding of biological invasions(14). Identifying and publicising the animal welfare impacts of biological invasions may forge collaborations between conservation science and animal welfare researchers that result in a more holistic understanding of the impacts of biological invasions on both biodiversity and animal welfare. Furthermore, although the biodiversity impacts of some alien species provide a compelling reason to develop rigorous biosecurity protocols to prevent their introduction, biosecurity remains poor in many regions(15). Identifying and publicising the biodiversity impacts of biological invasions justifies actions to improve biosecurity(16); identifying and publicising the animal welfare impacts of biological invasions may achieve the same outcome, particularly as there is broad public support for animal welfare(17).”*~~

(18) The paper does not provide a clear rationale for why welfare impacts should be studied separately from established biodiversity impacts. If welfare outcomes map directly onto fitness-reducing outcomes (injury, illness, death), then what is added by invoking welfare as a separate axis?

(18.1) Response: Thank you for pointing this out – we have added the following paragraph to the introduction of the main manuscript (**lines 86 – 105**) to justify the need for AWICIS:

“This framework is needed because the animal welfare impacts of biological invasions are a different type of impact to the biodiversity impacts of biological invasions (as assessed using the EICAT framework)(21, 26) – animal welfare impacts affect the mental and physical state of individual animals(4); biodiversity impacts affect the survival of entire species (their ability to reproduce and maintain viable populations). Furthermore, although the mechanisms through which biological invasions cause animal welfare and biodiversity impacts may often be the same (e.g., predation of a native species by an introduced species, which can cause declines in the population of the native species or even its extinction(27), and also the welfare of an individual of that species, which may suffer physically and emotionally whilst being preyed on(28)), the severity of the animal welfare and

biodiversity impacts caused through the same mechanism may not be congruent. For example, introduced parasites may not cause species extinctions (a severe biodiversity impact), because as a species' population declines as a result of parasitism, transmission of the parasite is reduced to low densities(29) – however, introduced parasites can cause severe suffering of individual host animals(15). Therefore, AWICIS may provide information on the severity of animal welfare impacts caused by biological invasions that cannot be obtained using existing frameworks such as EICAT. Finally, a dedicated animal welfare framework is required because biological invasions can affect the welfare of both native and introduced species, but existing frameworks that assess the impacts of biological invasions on non-human organisms (including EICAT(21, 26)) have been designed solely to assess impacts on native species (but not introduced species)."

(19) Methods

The impact mechanisms listed in Table 1 are not always mechanisms per se. For example, “structural changes to ecosystem” and “chemical changes to ecosystem” are environmental conditions that may enable welfare impacts, but are not direct mechanisms in the sense that predation or parasitism are. The list blurs causal chains, weakening analytical clarity.

(19.1) Response: The mechanisms we use for AWICIS are those used for the published EICAT framework. Some of these mechanisms cause direct biodiversity and animal welfare impacts; some cause indirect biodiversity and animal welfare impacts. For example, biodiversity impacts assessed under EICAT that are associated with the mechanism ‘chemical changes to ecosystem’ are not always direct. Introduced waterfowl can change the chemical composition of waterbodies (through nutrient enrichment from their droppings), leading to eutrophication and hence poor conditions for the survival of native species such as fish. This indirect biodiversity impact is also an indirect animal welfare impact, as fish suffocate as a result of eutrophication. As EICAT is a published framework that was developed through a long consultation process we feel it is sensible to adopt these mechanisms – though we do acknowledge the point made by the reviewer that some mechanisms are direct and some are not.

(19.2) Response: Reviewer 4 has provided an example of an indirect animal welfare impact for ‘chemical changes to ecosystem’ (as we did not find any during our literature review) (see Reviewer 4 comment below – point 42). This example demonstrates that leachate from an invasive shrub can cause risk-prone behaviour in a larval amphibian, increasing its risk of predation (<https://doi.org/10.1093/beheco/art121>). We have updated the Fig. S2 (Supporting

Information at line 101) and Table 1 in the main manuscript (at line 140) to include this example.

(19.3) Response: Reviewer 4 noted that for these complex impacts, we need to encourage clear reporting of the entire causal chain (point 40). We have included a paragraph on this in the guidelines (lines 353 – 363 in the Supporting Information):

“4.3 Describing the full causal chain associated with an impact

*Some welfare impacts are direct, such as predation of one animal by another – the processes associated with direct impacts tend to be relatively intuitive and straightforward to document. However, other welfare impacts caused by biological invasions are indirect (e.g., changes to the structure of vegetation communities at the landscape scale, caused by the introduction of cheatgrass (*Bromus tectorum*) to Western USA – because cheatgrass is flammable it increases the size of wildfires, and in turn the number of individual animals that they injure)(75). In such cases, it is important to describe the full causal chain leading to impact, in order to clearly demonstrate how welfare impacts on animals arise as a result of a biological invasion. The AWICIS assessment template (Section 7 of these guidelines) includes the requirement to describe the causal chain associated with welfare impacts.”*

(19.4) Response: We have also produced an assessment template which includes sections that must be filled out to: (i) note whether an impact is direct or indirect, and (ii) describe the complete causal chain associated with an impact (columns X and Y in the template - <https://figshare.com/s/29a0492309398e111c18>).

(19.5) Response: We note that in the example described above (point 19.2) there are two linked mechanisms associated with the impact – the first is the chemical change to the ecosystem, and the second is predation. As such we have also included sections in the template to report a second impact mechanism (if relevant for an indirect impact) (column AA in the template - <https://figshare.com/s/29a0492309398e111c18>).

(19.6) Response: We acknowledge that having both indirect and direct mechanisms in the framework does add complexity, but we feel it is better to identify and report all types of impact (both direct and indirect). We feel that being clear about the distinction between direct and indirect mechanisms may help those using the framework. As such, we have improved the descriptions of the indirect impact mechanisms (structural, chemical and physical

impacts on the environment) in Table 1 (main manuscript, at line 140) and we have updated Fig. S2 (Supporting Information, at line 101) to indicate which mechanisms are direct and which are indirect (by adding text to the edge of the chart).

(20) The framework does not define clear, objective criteria for distinguishing severity levels. Terms like “short-term suffering” or “novel suffering” are not operationalized in a way that different assessors could reliably apply. The approach risks being subjective and inconsistent.

(20.1) Response: Response: We use “short-term” and “prolonged” as a measure to distinguish impact severity. It enables distinctions to be made between level (ii) (short-term) and (iii) (prolonged) impacts, and between level (iv) (short-term) and (v) (prolonged) impacts. For many welfare impacts it should be possible to make the distinction between short-term and prolonged. For example, during our assessment, we found many predation impacts associated with introduced raptors (predatory birds) to be short-term, but we were able to assess most predation impacts caused by introduced ants as prolonged. More broadly, for example, we would expect many competition, disease and parasitism impacts (caused by biological invasions in general) to be categorised as prolonged.

(20.2) Response: We provided criteria for short term and prolonged impacts in the guidelines (in Section 3.2 (lines 195 – 204) and Table S2 at line 241). We should have included information on this in the main manuscript as well – we have added a short section to the introduction (lines 172 – 178) where we provide a brief overview of the framework. We have provided the text here for reference:

“When assessing severity, AWICIS provides guidance to determine whether impacts are short-term or prolonged (Supporting Information, Appendix A, Section 3). This enables distinctions to be made between level (ii) and (iii) impacts, and between level (iv) and (v) impacts (Supporting Information, Appendix A, Table S2). AWICIS stipulates an approximate duration of < 1 hour for short-term impacts, with any impacts lasting longer than 1 hour being categorised as prolonged. Intermittent impacts (e.g., competition impacts) that occur over long periods (> 1 hour) but not constantly should be categorised as prolonged.”

(20.3) Response: It may not always be possible to confidently determine the duration of an impact – and hence there may be some uncertainty when categorising it as short-term or prolonged. This is why each AWICIS assessment is also given a confidence score of ‘low’,

'medium' or 'high' (and the assessor also must provide a brief written explanation regarding the confidence score, to acknowledge and explain any uncertainty). Hence, confidence levels can be used to acknowledge uncertainty regarding impact duration. The published EICAT framework uses confidence scores to deal with uncertainty in biodiversity impact assessments.

(20.4) Response: Regarding 'novel suffering', the reviewer has encouraged us to be more precise in the way that we use terms in our manuscript and Supporting Information. We have edited the manuscript to remove the term 'novel' (which we used four times) – we felt this term was vague, and its meaning could be misinterpreted by the reader. Table 2 in the main manuscript (at line 170) which describes the five impact severity categories no longer uses the term 'novel' for level (iv) or (v) impacts. We have also removed the term novel from the Supporting Information.

(21) The reliance on literature-based biodiversity impact data is problematic. Most invasion-related literature is biased toward well-studied taxa, dramatic impacts, and high-income countries. The authors acknowledge this but continue to draw conclusions that generalize beyond the available data.

(21.1) Response: We agree that data on the impacts of biological invasions is skewed (taxonomically, geographically and towards more severe impacts). We do clearly acknowledge this in the main manuscript (lines 371 – 383). By demonstrating that we lack data on animal welfare impacts our study may encourage research to address this issue.

(21.2) Response: We acknowledge that some of the conclusions that we have drawn in our manuscript generalise, and at times are tangential to the main message of the manuscript. We have thoroughly edited the discussion which is now more concise (reduced from 3115 words to 2046 words). For example, we deleted the entire section on animal welfare impacts associated with failed introductions. We have also divided the discussion into sections with headings, as suggested by the reviewer in point 12 above. We also begin the discussion with a section on limitations to acknowledge the issue of data availability at the start (at line 359). The final lines of this paragraph (lines 380 – 383) set the overall tone for the rest of the discussion, which is much more cautious:

“Hence, absence of evidence on impacts should not be considered as evidence of absence of impacts – just because we did not identify welfare impacts for certain taxa

or regions does not mean they do not occur. The conclusions we draw in the following sections are based on the data that are available.”

(22) The treatment of impacts on introduced animals (e.g., two non-native species competing) adds conceptual confusion. What is the ecological or ethical relevance of these interactions within the context of a conservation or management-focused framework?

(22.1) Response: The purpose of the framework is to assess the animal welfare impacts of biological invasions. Affected animals may be native, but they may also be introduced. So we do think it's important to consider and assess welfare impacts on any animal regardless of its status. We think the text we added to the introduction about the benefits of the framework for biodiversity conservation may have been confusing – as suggested by the reviewer in an earlier comment, we have removed this text.

(23) The absence of physiological indicators in most cases is treated as a data availability issue, but it may in fact reflect a deeper problem: that many “welfare impacts” inferred here are not readily observable or measurable, casting doubt on whether this framework captures real, meaningful animal experiences.

(23.1) Response: Using physiological indicators is well-established in animal welfare science – many behavioural and physiological welfare indicators are described and discussed in animal welfare textbooks and papers (e.g., Fraser and Broom, 1996 (<https://www.cabidigitallibrary.org/doi/full/10.5555/19962214501>); Mendl *et al.*, 2017 (<https://psycnet.apa.org/doiLanding?doi=10.1037%2F0000012-035>); Appleby *et al.*, 2018 (<https://www.cabidigitallibrary.org/doi/book/10.1079/9781786390202.0000>)). However, it is a challenge for the relatively new and evolving field of wild animal welfare where remote data collection methods are likely to be needed (see <https://doi.org/10.1111/brv.13009>). Using the framework to demonstrate that we lack physiological impact data will hopefully encourage more research in this area. Animal welfare impacts caused by biological invasions have been identified using physiological indicators, so it is possible to do so (we list several examples of studies undertaken on this topic in Table S4, Supporting Information) However, it seems that such research techniques have yet to be broadly adopted or applied to biological invasions.

(24) Results

The figures (especially Figures 1 and 3) are excellent, though some formatting overlaps (e.g., Figure 1c) should be corrected. The visualization of mechanisms, domains, and impact severity is helpful.

(24.1) Response: Thank you – we have amended two figures to address this issue (Fig. 1c and Fig. 3c).

(25) The method of counting “impacts” based on published studies raises several concerns:
1) It treats each publication as equally reliable, regardless of methodology.

(25.1) Response: We acknowledge that there is likely to be some variation in quality across these publications. To control for reliability in terms of the research methods adopted (and research quality more broadly) we only included articles in our review that were published. As these articles have been published, the research they describe is likely to have been planned, methodically reported and peer reviewed (in comparison to unpublished studies). The methods and research ideas must also have been considered worthy of publication (by researchers and editors). Other studies that have used this approach (counting impacts using published studies) have been published in reputable scientific journals with rigorous review standards, such as *Ecography* (<https://doi.org/10.1111/ecog.05000>).

(26) 2) It assumes geographic and taxonomic coverage is unbiased, which is demonstrably false.
3) It may conflate absence of evidence with evidence of absence, particularly in low-income regions or underrepresented taxa.

(26.1) Response: As discussed previously, we do acknowledge that bias exists in research for invasion biology. However, many important studies on the distribution and impacts of introduced species have been published in reputable journals using data on introduced species that is biased in this manner. Indeed, the recent IPBES alien species assessment (<https://www.ipbes.net/ias>) included analysis based on datasets that are skewed towards more developed regions and more severe impacts (including global datasets on the biodiversity impacts of introduced birds provided by one of the authors of this manuscript (Tom Evans)). Tom is a contributing author of Chapter 4 of the IPBES assessment which assessed introduced bird impacts.

(26.2) Response: As stated previously, we have toned down the language regarding some of the conclusions we make based on the results of our analysis. We acknowledge at the

start of the discussion that our datasets are likely to be biased towards certain regions and types of impacts. We state that a lack of evidence on the welfare impacts of introduced species does not necessarily mean that such impacts do not occur (lines 371 – 383).

(26.3) Response: We think it's important to publish the data we have produced in our assessment in order to show that we are likely to lack information on certain types of animal welfare impacts and in certain regions. Demonstrating this bias may encourage more research on this topic (which we think is neglected when compared to the biodiversity and socio-economic impacts of introduced species).

(27) 4) The decision to merge severity levels (i–iii vs. iv–v) is pragmatic but analytically limiting. It obscures variation within lower-level impacts, which may still be biologically or ethically relevant.

(27.1) Response: Grouping the categories as 'less severe' (i, ii and iii), and 'more severe' (iv and v), allows for a key distinction between these two groups. 'Less severe' impacts are those that are no more severe than those the affected animal experiences in the absence of biological invasions. 'More severe' impacts are those that are more severe than those the animal is affected by in the absence of the biological invasions. Therefore, whilst we acknowledge there is some loss of analytical nuance by merging impact severity categories, we do believe it enables broad patterns in impact severity to be identified using this key distinction. Reviewer 4 is supportive of this approach (see point 43).

(27.2) Response: Regarding 'less severe' impacts (i – iii), for ant invasions, merging them was appropriate, as there were very few data points in two 'less severe' impact categories: (i), $n = 2$; (ii), $n = 1$. For bird invasions, there were also many categories of interest where there are very few data points for 'less severe' impacts. For example, no predation impacts were categorised as (i), and no hybridisation impacts were categorised as (ii) or (iii). Therefore, if we were to split out the 'less severe' categories (i – iii) and analyse them separately, we would have many '0' values for analysis in our contingency tables. Furthermore, as we had grouped the ant impact dataset into 'less severe' (i – iii) and 'more severe' (iv and v), it was appropriate to be consistent and do the same for the bird impact dataset.

(28) 5) The high frequency of behavioral indicators, relative to physiological or direct observational data, suggests a potential over-reliance on interpretation. Many reported

behaviors (e.g., nest abandonment, listlessness) could have multiple causes, not all linked to invasive species.

(28.1) Response: The studies we extracted impact data from were all published in journals and directly concerned with the impacts of introduced species. Hence, we assume that the quality of these studies is good (as they have been published). It is possible that the researchers that published these studies incorrectly interpreted the behaviour of the study species. If this was the case, then presumably their assessment of the causes of the biodiversity impacts they reported would also be wrong (just as our assessment of animal welfare impacts would be). For example, Strubbe *et al.* (2009) identified nest competition with introduced rose-ringed parakeets (*Psittacula krameri*) as a reason for native nuthatches (*Sitta europaea*) abandoning their nests in Belgium (<https://doi.org/10.1016/j.biocon.2009.02.026>). There may have been other causes for this behaviour, but their research led them to believe that the introduced parrot species was the cause. If their assessment is wrong then their conclusion regarding impacts on biodiversity are wrong, just as our conclusions for animal welfare are.

(28.2) Response: AWICIS includes confidence categories ('low', 'medium' or 'high') to indicate the level of confidence an assessor has in their assessment of impact severity. These can be used when considering the effect of confounding factors on the behaviour of an animal in response to a biological invasion.

(29) Discussion

The discussion is too long and poorly structured, mixing findings and speculation without a clear through-line. A more structured format (e.g., "Findings for Birds", "Findings for Ants", "Conceptual Limitations", "Policy Implications") would enhance readability and focus.

(29.1) Response: We have thoroughly revised the discussion implementing the structure suggested by the reviewer, including headings. We have also edited the discussion to reduce its length (by approximately one-third). We also added headings to the introduction.

(30) Some language is excessively rhetorical or anthropomorphic. Phrases like "death is both extremely painful and prolonged" or "panic behavior" need to be grounded in evidence, not intuition. Avoid emotionally loaded terminology unless backed by physiological or experimental data.

(30.1) Response: We have reviewed the language to remove emotional terminology, including in:

- Fig. S2 in the Supporting Information (at line 101) which describes the impact mechanisms with examples.
- the detailed example of parasitism in the introduction (lines 51 – 58 of the main manuscript).
- the description of the indicators in the results (lines 261 – 275 and lines 316 – 339 of the main manuscript).

(30.2) Response: We have deleted the phrase ‘death is both extremely painful and prolonged’. Please note the term ‘panic behaviour’ was taken directly from the article that reported the impact

(www.jenniferelainesmith.com/uploads/3/8/4/1/38419411/smith_et_al_2007_novel_predator-prey_interactions.pdf). We have changed this term to ‘vigorous avoidance and defensive behaviour’ (lines 329 and 434 of the main manuscript). Similarly, the term ‘scalped’ was taken directly from the article describing the impact (doi:10.1017/S0954102015000486). We have deleted this term in Fig. S2 (Supporting Information at line 101). We agree that the manuscript should report and discuss results without emotion and rhetoric – we believe it now does that. Please note that we do still use the term ‘suffering’ – as we think it is important to convey the fact that animal welfare impacts can be unpleasant both emotionally and physically, and this term is used in articles on animal welfare.

(31) The repeated emphasis on islands as hotspots of welfare impacts mimics known biodiversity patterns, but the authors don’t explain why welfare should behave similarly. If predation and competition are more intense on islands, say so—but don’t assume a causal link without justification.

(31.1) Response: Welfare impacts on islands are often more severe because the native species on these islands do not suffer from welfare impacts until invasive species are introduced. We discuss this at lines 391 – 397 of the main manuscript. We note this applies to many introduced bird impacts – but not introduced ants, which have severe welfare impacts in most places they are introduced (islands and mainland locations). We discuss this at lines 418 – 428 of the main manuscript.

(32) The assertion that AWICIS can inform management decisions or conservation planning is premature. The data are too uneven, the categories too subjective, and the framework too

new to justify policy relevance at this stage.

(32.1) Response: Point taken – we have removed all of this text. We have included a dedicated section in the discussion on policy implications and future directions for research (lines 482 – 513 of the main manuscript). Instead of discussing the ways in which AWICIS may be used to influence management decisions (we deleted this text), we have instead focussed on encouraging use of the framework to address data gaps.

(33) The final paragraphs drift into speculative ethical territory. The suggestion that suffering during failed introductions should be considered when crafting animal welfare law may be ideologically driven rather than scientifically justified.

(33.1) Response: We have deleted the text discussing animal welfare law. However, we do think that there are important observations regarding the lack of data on impacts on introduced species, and we do discuss this (lines 505 – 513 in the main manuscript). However, this section is now concise and sticks to broad points regarding the need to consider welfare impacts on introduced species (not only native species).

Reviewer #4

(34) Overview

The authors present a framework (AWICIS) for quantifying the impacts of biological invasions on animal welfare. This is an important problem which has not, to my knowledge, been covered by previous frameworks of invasive species impact assessment - for this reason, I think that the manuscript is a significant contribution to the field.

(35) To categorise the welfare impacts of biological invasions, AWICIS adapts parts of existing frameworks for assessing the biodiversity impacts of invasive species (EICAT) and for animal welfare assessment (the Five Domains model), but is novel in the way that it combines these frameworks. In my view, the correspondence between AWICIS and EICAT is a strength of the manuscript, as it could facilitate the simultaneous application of both frameworks to a given biological invasion.

(36) The next part of the framework, which quantifies impact severity by comparing impacts caused by invasive species with those caused by native species of the same class through the same mechanism (and also considers impact duration) is novel, practical to implement, and is a sensible comparison to make.

(37) The authors also present a case study in which they apply their framework to welfare impacts of invasive birds and ants, based on analysing existing literature.

(38) Overall, the authors present a highly detailed and thorough manuscript.

(38.1) Response: Thank you for these comments. We hope that AWICIS will be used in the future to compare the animal welfare impacts of biological invasions with their biodiversity impacts assessed using EICAT (and also the socio-economic impacts of biological invasions assessed using SEICAT).

(39) I have some comments on both the framework itself, and on the case study, which I outline below.

(40) Comments on the AWICIS framework

1. Evidencing the whole causal chain linking invasions to welfare. Providing evidence to demonstrate the welfare impacts of invasive species is the 4th stage of the AWICIS framework (Fig. S1), and the authors provide numerous examples in the supplementary tables and in the main text. However, what I think is missing in places is an explicit consideration of the entire causal chain which links the biological invasion to the welfare impact. For example, in Table S15 the authors present an example in which cheatgrass impacts the welfare of wapiti by increasing the intensity of wildfires. In this example, the indicator is given as “Physical. Animals injured and killed by wildfires”. However, the evidence required to assign an impact severity of (v) is actually that cheatgrass increases the intensity of wildfires (otherwise, the impact could be graded (iii) if wildfires were of a similar intensity in the absence of cheatgrass). In other words, the causal chain is: cheatgrass \diamond wildfires \diamond welfare of wapiti, but only the latter link is evidenced. Therefore, I think it would be a good idea for the authors to discuss the importance of evidencing the full causal chain (especially in the section starting line 258 of the supplement), and to consider adding this requirement to the framework (Fig. S1).

(40.1) Response: We think this comment is particularly important for ‘indirect’ impacts. We have included the following paragraph in the guidelines (lines 353 – 363 of the Supporting Information) – both to emphasise that some impacts are indirect, and to encourage full reporting of the causal chain.

“4.3 Describing the full causal chain associated with an impact

*Some welfare impacts are direct, such as predation of one animal by another – the processes associated with direct impacts tend to be relatively intuitive and straightforward to document. However, other welfare impacts caused by biological invasions are indirect (e.g., changes to the structure of vegetation communities at the landscape scale, caused by the introduction of cheatgrass (*Bromus tectorum*) to Western USA – because cheatgrass is flammable it increases the size of wildfires, and in turn the number of individual animals that they injure)(75). In such cases, it is important to describe the full causal chain leading to impact, in order to clearly demonstrate how welfare impacts on animals arise as a result of a biological invasion. The AWICIS assessment template (Section 7 of these guidelines) includes the requirement to describe the causal chain associated with welfare impacts.”*

(40.2) Response: We have also produced an assessment template (described in Section 7 of the guidelines in the Supporting Information – lines 416 – 423), which will help assessors to consider all aspects of the framework when assessing impacts. It includes specific requirements to categorise impacts as direct or indirect and to report the causal chain (columns X and Y in the template). It also includes several impact examples (these were the examples included in Table S15 in the previous version of the Supporting Information, which we have deleted). The assessment template has been uploaded to a repository where it can be periodically updated, and the latest version made available for assessments:

<https://figshare.com/s/29a0492309398e111c18>

(40.3) Response: We have updated Fig. S1 in the Supporting Information (at line 73 – see Section 4. ‘Providing evidence’) to include the requirement to describe the full causal chain leading to an impact. We have updated Fig. S2 in the Supporting Information (at line 101) to make it clear that some impacts are direct and others indirect (by adding text to the edge of the chart).

(41) 2. Consideration of impact prevalence / extent. In my view, it is not a good idea to exclude changes to impact frequency from the consideration of “prevalence” (as stated in Table S1). For example, a native prey animal may have native predators, but predation by an invasive predator may be far more prevalent (due to e.g., native prey naïveté, or higher abundance of invasive predators than native predators). Consequently, a greater number of native prey animals may experience welfare impacts than they would if the invasive predator was absent. However, from my understanding of the framework, if the invasive predator

does not increase the spatial extent of predation, then the impact would only be assigned severity ii or iii, regardless of the number of additional native animals which are predated. Conversely, if the invasive predator increases the spatial extent of predation, then the impact could be assigned to category iv or v, even if the predation rate is low – to me, this seems inconsistent. A similar argument could be made about some of the other impact mechanisms, such as disease – would additional spread by the invasive species not impact the welfare of native individuals which would otherwise be unaffected? In my view, it would be better to allow for “increased prevalence” to include cases where there is clear evidence that biological invasion results in more frequent impacts, and for these cases to be assigned to the higher impact categories.

(41.1) Response: Thank you for this comment – we have updated the framework and added an example to Table S3 in the Supporting Information (at line 265) for an impact associated with increased frequency (introduced Australian brushtail possums (*Trichosurus vulpecula*), which are the most important (but not the only) maintenance host for bovine tuberculosis in New Zealand, providing interface for transmission between livestock and wild forest animals).

(41.2) Response: We hesitated to include frequency in the previous version of the manuscript because the framework measures relative changes to an individual’s welfare that result from a biological invasion, and even though an impact may be more frequent, it may not necessarily be more severe. If an impact is more severe, it qualifies as being (iv) or (v) simply by that fact (not because it’s more frequent). However, we acknowledge that increasing frequency may also dramatically increase the chances of an individual being affected by an impact when it otherwise wouldn’t be – so we now agree that frequency should be included as a criterion for more severe (iv and v) impacts.

(42) 3. Examples of different impact mechanisms (Table 1 and Figure S2). It is not very clear how some of these mechanisms are defined (e.g., what is the difference between a “physical change” and a “structural change”, or between a “chemical change” and “poisoning/toxicity”?). I think it would be good to expand on these definitions (at least in the table if there isn’t room in the figure). Also, for some of the impact mechanisms, the authors did not find any examples of the mechanism in the literature. I have a couple of suggestions for the authors to consider:

- Physical changes to ecosystem: Carter et al. (2014). Exotic invasive plants alter thermal regimes: implications for management using a case study of a native ectotherm. *Functional Ecology*. <https://doi.org/10.1111/1365-2435.12374>

- Chemical changes to ecosystem: Hickman & Watling (2014). Leachates from an invasive shrub causes risk-prone behavior in a larval amphibian. *Behavioral Ecology*. <https://doi.org/10.1093/beheco/art121>

(42.1) Response: Thank you for the additional mechanism examples. We have improved the definitions in Table 1 (main manuscript, at line 140) for chemical, structural and physical changes to ecosystems (impact mechanisms 8, 9 and 10, respectively). To provide context, we have also provided an impact example for each of these mechanisms.

(42.2) Response: We have added examples to the mechanisms figure in the Supporting Information (Fig. S2, at line 101). We have further amended this figure to make a distinction between these more complex indirect mechanisms (8 to 11) and the direct mechanisms (mechanisms 1 to 7). To do this we have added text to the edge of the chart to denote direct and indirect mechanisms.

(43) Comments on the case study

1. Statistical analyses. In their case study, the authors use contingency table tests to examine how the severity of welfare impacts (grouped into “more severe” and “less severe” impacts, which I think is fine) varies according to impact mechanism, location etc. However, there are some issues with the way these are reported. First, it is unclear which test was actually used – the table legends in Appendix B state that Fisher’s exact test was used, but the table reports χ^2 statistics, which Fisher’s exact test doesn’t use.

(43.1) Response: We should have included this information – apologies. It is now included in both the main manuscript (line 609) and the Supporting Information (at the start of Appendix C, line 531). We have used exact tests and the FunChisq package in R.

(44) Second, both Fisher’s exact test and the χ^2 test assume that the observations are independent, but this assumption could conceivably be violated (e.g., if one invasion generates impacts across multiple mechanisms, across multiple domains of animal welfare, or across multiple native species). Therefore, the authors should consider the assumptions of their statistical tests, and justify them in the methods section.

(44.1) Response: The counts are of each impact described in an article on a biological invasion. If an article described different types of impacts, these were analysed as separate impacts. Each impact was specific to a location, mechanism and affected animal. The analysis was used to identify variation in impact severity – each impact was categorised as being either ‘less severe’ or ‘more severe’. No impact was categorised as being in both ‘less severe’ and ‘more severe’ categories. We have updated the text in the main manuscript to make this clear (lines 608 – 616).

(44.2) Response: Reviewer 1 was concerned with the number of tests and small cell values (see point 8). We have repeated our response to this comment below. We have updated the methods section accordingly.

“(8) I am not enamoured by the large number of contingency table tests, or the small size of the cell counts in many cases. I suspect that a lot of basic assumptions are being violated across these tests.

(8.1) Response: We have critically reviewed our approach to the contingency table tests. There were 11 in the submitted manuscript – on reflection, five of these tests were not particularly useful, so we have deleted them. The deleted tests were as follows:

- two tests analysing the distribution of impact severity across continents (one test for ants and one test for birds). We do not think examining patterns at this broad spatial scale provides useful insights other than to reinforce patterns that we have already shown using the global map of impacts.
- two tests describing the distribution of impact severity across domains (one test for ants and one test for birds). We have already tested impact severity across impact mechanisms – the links between the domains and the mechanisms are shown in Fig. 1 and Fig. 3. In short, the impact mechanism ‘predation’ tended to be more severe for both bird and ant invasions – Fig. 1 and Fig. 3 clearly display which domains are associated with predation.
- one test examining the distribution of impact severity across status of affected animal (either native or introduced) for bird invasions. There were no ‘more severe’ impacts reported for introduced animals affected by these invasions (see table below). Therefore, we think it is more appropriate (and straightforward) to just describe this result (lines 280 – 282 in the main manuscript).

	Introduced	Native
Less severe	305	472
More severe	0	79

- *Note: we did not test impact severity across status of affected animal (native or introduced) for ant invasions in the original manuscript as there no reports of wild, introduced animals being affected by ant invasions.*

(8.2) Response: *The manuscript now includes six contingency table tests in total (three each for bird and ant invasions, respectively) (Appendix B). They are presented more concisely in the supporting information (three tests to a single A4 portrait page) rather than spanning several landscape pages as in the originally submitted supporting information. As such it is easier to digest the information they contain.*

(8.3) Response: *Following the advice in the Handbook for Biological Statistics (<https://www.biostathandbook.com/small.html>) we pooled some categories to reduce the number of cells with small values (see ‘Pooling’ section in <https://www.biostathandbook.com/small.html>). We have provided details on the pooled categories below each contingency table test in the Supporting Information (Appendix B). Also following advice in the Handbook for Biological Statistics, we used exact tests, as all our tables had total sample sizes of less than 1000 (see ‘Recommendation’ section in <https://www.biostathandbook.com/small.html>). We also used exact tests because after pooling, three cells in tests for ant invasions did contain values < 5 (but none had 0 values), and 5 is considered a ‘cut-off’ for using exact tests (see ‘The problem with small numbers’ section in <https://www.biostathandbook.com/small.html>). We have described this in the main manuscript (lines 635– 639). We undertook the analysis using the FunChisq package in R – we have included this information in the Supporting Information (at the start of Appendix C, line 531) and the main manuscript (line 609). The results of the revised contingency table analysis are similar to those produced by the original analysis – they do not change the broad conclusions presented in the discussion.”*

(45) Third, I could not find information how the tests were actually implemented (e.g., was a specific software package used, or were the calculations done by hand?). This detail should be provided in the methods section.

(45.1) Response: We used the FunChisq package – we have added this information to the main manuscript (line 609) and the Supporting Information (at the start of Appendix C, line 531).

(46) Finally, the “statistics” section of the reporting summary checklist seems to contain some items where “n/a” was ticked when the authors could have ticked “confirmed” (e.g., the authors provide the full output from the contingency table tests in the supplementary tables, so could tick the item beginning “for null hypothesis testing...”).

(46.1) Response: Thank you – we have updated the checklist.

(47) Other comments

- The justification for applying the framework to birds and ants – that they are dissimilar taxa – is reasonable. However, readers may appreciate some brief context (either in the intro or the methods) to illustrate the importance of bird/ant invasions worldwide (e.g., what proportion of documented invasive species are birds/ants?).

(47.1) Response: We have added a paragraph to the methods (lines 520 – 530 in the main manuscript) to give some context to the assessment, describing the distribution and biodiversity impacts of introduced ants and birds.

*“We used AWICIS to assess the animal welfare impacts of bird and ant invasions. Birds have been introduced to novel environments for thousands of years(61) – introduced bird species are now established in most of the world’s bioregions(62) including on many small islands(43). The number of established introduced bird species continues to rise(1), and some are reported to have severe impacts on biodiversity(22) – for example, predation by introduced birds is likely to have contributed to the extinction of at least four native bird species on small islands(43). Ants have also been introduced to many regions of the world – there are at least 520 introduced species that are established worldwide(56). Three of these species are amongst the ten most widespread invasive insects(1). Ants are often considered to be amongst the most damaging introduced species – 17 ant species are known to have severe impacts on native biodiversity(56), including the yellow crazy ant (*Anoplolepis gracilipes*) which through predation has severely reduced the size of the red crab (*Gecarcoidea natalis*) population on Christmas Island(63).”*

(48) - Lines 204-295 – to me this sentence implies that ants have a broader range of impacts despite the birds being more taxonomically diverse, when in fact there are at least several thousand more species of ants than birds – consider rephrasing?

(48.1) Response: Thank you – we agree that this text was confusing and have revised it as follows (lines 418 – 421 in the main manuscript):

“The data that are available suggests that a greater range of taxa are vulnerable to severe welfare impacts caused by ant invasions when compared to bird invasions. The ant invasions in our dataset also consistently cause severe welfare impacts wherever they occur, whereas the bird invasions rarely do.”

(49) - Lines 596 and 599 – shouldn't islands be defined as “< 20,000 km²”, rather than “> 20,000 km²” (i.e., large ‘islands’ like the UK do not count)?

(49.1) Response: Yes – this was an error, which we have corrected as suggested (lines 619 and 622 in the main manuscript).

(50) - Line 773 – the link to the reference doesn't work

(50.1) Response: We have checked all the links in the document.

(51) - Supplement lines 325-326 – I think the probabilities assigned to the low/medium/high confidence categories are unnecessary, and it is unclear how these probabilities could ever be objectively determined. In my opinion it is better just to have the low/medium/high labels, with the written explanation as the framework already requires.

(51.1) Response: On reflection we agree with this comment – we have removed the probability requirements for the confidence categories (lines 408 – 410 in the Supporting Information).

(52) - Table S3 – “livestock salivating” is provided as an example of an indicator, but from this table alone it's unclear why this would be indicative of welfare impacts. Table S15 clarifies that the livestock exhibit excessive salivation, so the wording in Table S3 should probably be changed to “livestock excessively salivating” or similar.

(52.1) Response: Agreed – we have updated the description in the table (now Table S4, at line 350 in the Supporting Information). We use ‘livestock excessively salivating’ as suggested.

(53) - Table S15 – in the South Africa / Marion Island example, the “impact mechanism” (5th column) refers to owls, when it should refer to mice.

(53.1) Response: Thank you – we have amended this text. These examples are now in the assessment template (<https://figshare.com/s/29a0492309398e111c18>) rather than Table S15, which has been deleted.

(53.2) Response: Each line of the template reports welfare impacts for one animal only (unlike in Table S15 which has been deleted). This is because we felt the reporting process became rather complex in Table S15, and could be confusing to assessors. So the template treats each welfare affected animal in exactly the same manner whether it be native, introduced, wild or domesticated – each is afforded a discrete row. Hence impacts of an introduced owl on a native bird would be reported in one row, and impacts of the native bird on the introduced owl on another. We have provided examples to demonstrate this (e.g., the first two rows, which are for impacts on cane toads caused by northern quolls, and impacts on northern quolls caused by cane toads).

Response to comments

We thank Reviewer #5 for reading our manuscript to check our responses to previous reviewer comments. We have responded to the additional comments provided by the reviewer in blue text below. We have provided line number references for changes made to the main manuscript.

Reviewer #4

I am happy to report that all of my comments from the first round of review have been addressed. I have no further comments to add on the revised version of the manuscript.

Reviewer #5

Thanks for the opportunity to review this interesting manuscript. As I was asked to, I have checked the responses of the authors to the reviewers, and they seem to me to be exhaustive and convincing.

I only add a few comments here:

(1) One point that could be added is the sufferance of non-native individuals due to their management (e.g. trapping, killing – there are some data published on this). Although unavoidable, this is surely something that occurs. In this regard, cautious writing is also needed to avoid biological invasions negationists and animalists using this argument to stop management efforts; I would stress the need to minimize the number of suffering animals via early and rapid management, so that “only” a few animals will suffer, compared to what happens over time when interventions are late and populations became too large.

(1.1) Response: We agree that the animal welfare impacts caused by the management of biological invasions is an important topic. However, the framework we have developed is designed to assess impacts resulting from interactions between animals (not interactions between humans and animals). The impacts caused by the management of alien species is a separate topic – trying to integrate these types of impacts into the framework would be difficult. We have added a section to the AWICIS guidelines (Section 1.6, page 5, Supplementary Information) to address this issue. We refer to a manual produced by the IUCN which aims to improve welfare in the management of biological invasions:

Smith, K.G., *et al.* 2022. *A manual for the management of vertebrate invasive alien species of Union concern, incorporating animal welfare (No. 07). Technical report prepared for the European Commission within the framework of the contract.*

(1.2) Response: We also state that these types of welfare impacts fall outside of the scope of the AWICIS framework, and that AWICIS should not be used to try and assess them.

(2) Checking the more relevant literature, I found these two papers that I think should be included in the manuscript, given their very pertinent topic:

- Carneiro, L., Leroy, B., Capinha, C., Bradshaw, C. J., Bertolino, S., Catford, J. A., ... & Courchamp, F. (2025). Typology of the ecological impacts of biological invasions. *Trends in Ecology & Evolution*, 40(6), 563-574.
- Haubrock, P. J., Everts, T., Abreo, N. A. S., Bojko, J., Deklerck, V., Dickey, J. W., ... & Britton, J. R. (2025). The impacts of biological invasions. *Biological Reviews*.

(2.1) Response: Thank you for these references – we have added both to the introduction of the main manuscript (line 61). They help to strengthen our argument that the welfare impacts of biological invasions is a neglected research topic in comparison to their biodiversity impacts.